# Estimating genetic variability among diverse lentil collections through novel multivariate techniques

Syed Atiq Hussain[1], Muhammad Sajjad Iqbal[1]*, Muhammad Akbar[1], Noshia Arshad[1], Saba Munir[1], Muhammad Azhar Ali[1], Hajra Masood[1], Tahira Ahmad[1], Nazra Shaheen[1], Ayesha Tahir[2], Muhammad Ahson Khan[3], Muhammad Kashif Ilyas[3], Abdul Ghafoor[3]

1 Biodiversity Informatics, Genomics and Post Harvest Biology Laboratory, Department of Botany, University of Gujrat, Gujrat, Pakistan, 2 Departmet of Biosciences, COMSATS University, Islamabad, Pakistan, 3 National Agricultural Research Center (NARC), Islamabad, Pakistan

☯ These authors contributed equally to this work.

* drsajjad.iqbal@uog.edu.pk

**Data Availability Statement:** All relevant data are within the article and its Supporting Information files.

## Abstract

Lentil is an important food legume throughout the world and Pakistan stands at 18[th] position with 8,610 tons production from 17,457 hectares. It is rich in protein, carbohydrates, fat, fiber, and minerals that can potentially meet food security and malnutrition issues, particularly in South Asia. Two hundred and twenty lentil genotypes representing Pakistan (178), Syria (14), and the USA (22) including 6 from unknown origins were studied for yield, yield contributing traits, and cooking time (CT). Genotype 6122 (Pakistan) performed the best during both years with seed yield per plant (SY) 68±1.7 g, biological yield per plant (BY) 264±2.8 g, pod size (PS) 0.61±0.01 cm, number of seeds per pod (NSP) 2, cooking time (CT) 11 minutes, with no hard seed (HS). The genotypes 6122 (Pakistan) and 6042 (Syria) produced the highest BY, hence these have the potential to be an efficient source of fodder, particularly during extreme winter months. The genotypes 5698 (Pakistan) and 6015 (USA) were late in maturity during 2018–19 while 24783 and 5561 matured early in 2019. A minimum CT of 10 minutes was taken by the genotypes 6074 and 5745 of Pakistani origin. The lowest CT saves energy, time, and resources, keeps flavor, texture, and improves protein digestibility, hence the genotypes with minimum CT are recommended for developing better lentil cultivars. Pearson correlation matrix revealed significant association among several traits, especially SY with BY, PS, and NSP which suggests their use for the future crop improvement program. The PCA revealed a considerable reduction in components for the selection of suitable genotypes with desired traits that could be utilized for future lentil breeding. Structural Equational Model (SEM) for SY based on covariance studies indicated the perfect relationship among variables. Further, hierarchical cluster analysis establishes four clusters for 2017–18, whereas seven clusters for 2018–19. Cluster 4 of 2017–18 and cluster 5 of 2018–19 exhibited the genotypes with the best performance for most of the traits (SY, BY, PS, NSP, CT, and HS). Based on heritability; HSW, SY, BY, NSP were highly heritable, hence these traits are expected for selecting genotypes with genes of interest and for future lentil cultivars. In conclusion, 10 genotypes (5664, 5687, 6084, 6062, 6122, 6058, 6087,

**Funding:** There is no financial assistance for publication of this research article.

**Competing interests:** The authors have declared that no competing interests exist.

5689, 6042 and 6074) have been suggested to evaluate under multi-location environments for selection of the best one/s or could be utilized in hybridization in future lentil breeding programs.

## Introduction

Lentil (*Lens culinaris* Medik.) is an annual, herbaceous, cool-season, autogamous (self-pollinated), and diploid legume with a 4 Gbps genome size that was domesticated during the same era with wheat and barley in the Fertile Crescent [1,2]. It is one of the oldest dry legumes domesticated about 9,000 years ago in the Near East and subsequently spread throughout the Mediterranean Basin and Central Asia [3]. Lentil is the 4th major pulse crop in the world after common bean, pea, and chickpea. It includes seven taxa with four species such as *L. culinaris*, *L. nigricans*, *L. ervoides*, and *L. lamottei*, the only cultivated species (*L. culinaris*) has two varietal types, *i.e.*, macro-Sperma (large-seeded) and micro-Sperma (small-seeded) [4,5].

Lentil is being consumed due to its highest protein contents, vitamins, and micronutrients that are a vital part of human diet while vegetative parts are being used as fodder for livestock [6]. Protein in lentil seed is of high quality ranging from 22 to 35% and consisted of a large number of amino acids including lysine and isoleucine, hence it is a cheap source of protein in many regions of the world both for humans as well as for animals [7].

Additionally, lentil is a rich source of carbohydrates (59%), minerals (2%), and fiber with very little fat that has significant role to overcome malnutrition and deficiency of micronutrients in developing countries [8]. It is one of the best sources of iron [9] and medicinally it has low glycemic content, hence strongly suggested it for patients with diabetes, obesity, and heart maladies [9]. Lentil grains are mucilaginous and laxative recommended for treating constipation, stomach infections, smallpox, and ulcer [6]. The top ten lentil-producing countries are Canada, India, Turkey, the USA, Kazakhstan, Nepal, Australia, Russia, Bangladesh, and China [10].

Imperative causes of the low yield of lentils in Pakistan are mainly attributed towards cultivation on marginal lands with poor management practices, lack of high yielding varieties, adaptability of low yielding cultivars with narrow genetic base, biotic and abiotic stresses, [11,12]. Economically it is not feasible to reduce biotic and abiotic stresses by utilizing chemicals [13]. Alternatively, the best solution to overcome these problems is to develop biotic and abiotic resistance in crop plants that is possible through using broad-based genetic resources. Convetional breeding and modern bio-techniques could improve crops by pyramiding of genes of economic importance or transferring genes of interest in improved cultivar/s. Inadequate information of genetic variability and limitations of molecular techniques for genetic improvement is a major hindrance to harvest full potential of crop [13].

Several studies have been carried out worldwide to assess genetic diversity based on phenotypic evaluation in lentils as well other legumes in the past [3,7,11,14–19]. Germplasm repository is a valuable asset that holds a huge treasure of desirable genes for crop improvement, however it can only be accomplished through the use of knowledge on extent of genetic diversity [20]. Genetic diversity can be estimated on the basis of qualitative and quantitative traits, and, a large number of morphometric techniques have been used to measure and classify genotypes in various crops as reported such as mustard, peas, mungbean, alfalfa, cowpea, soybean, and lentil [21–26]. This study used several statistical techniques aimed to investigate genetic diversity in agromorphological traits for two consecutive years to unwind the hidden potential

of lentil genetic resources. Identified novel genotypes would be the base material for future lentil breeding programs, either through simple selection from multi-location testing or using in hybridization program.

## Material and methods

### Ethics statement on experimental research

The study was approved by the Institutional Committee through No. UOG/ASRB/Botany/03/15806. Accordingly, all of the current experimental research and field studies comply with relevant institutional, national, and international guidelines and legislation.

### Plant material, study site, research design, and crop cultivation

Two hundred and twenty lentil genotypes representing Pakistan, the USA, Syria and lines with unknown origins were obtained from the National Gene bank, National Agricultural Research Center (NARC), Islamabad, Pakistan (S1 Table). Germplasm was planted under field conditions of NARC in Islamabad, Pakistan located at 33.6701˚ N latitude and 73.1261˚ E longitude during October 2017 and harvested in April 2018, similarly, the same timing and procedure were followed in the next year 2018–19. Both qualitative and quantitative traits were recorded consecutively for two years (2017–18 and 2018–19). While meteorological data recorded on monthly basis was obtained from the experimental station (S1 Fig). Experiments were laid out in an augmented design in 11 blocks and each block consisted of 20 genotypes along with two check varieties, *viz*., Markaz 2009 and Punjab 2009 after every twenty lines [16]. Three rows of 5 m length in each bed were planted with 10 cm space between plants and 30 cm within rows. The trial was conducted under usual agricultural practices without applying any herbicide, fungicide, pesticide, or fertilizer so that the actual genetic diversity may be studied. The soil was homogenously fertile and analyzed in Soil Science Laboratory, NARC, Islamabad. The soil was Nabipur series, coarse loamy, mixed, hyperthermic, and udic ustochrepts. Recommended cultural practices were followed throughout the crop season [27].

**Data collection.** Ten healthy plants were randomly sampled from each genotype for recording of the data on a plant basis. Quantitative traits were recorded at foliage, reproductive and maturity stages including plant height (PH), number of pods per peduncle (NPP), lower pod height (LPH), pod size (PS), number of seeds per pod (NSP), and days to maturity (DM). Days to maturity were considered from the day of sowing to turning of 90% of pods into golden brown color. After harvesting, data on biological yield (BY), seed yield (SY), 100 seed weight (HSW), cooking time (CT), and hard seeds (HS) were recorded. Cooking time in minutes was recorded after boiling of 100 un-soaked seeds to softness in distill water.

For qualitative traits, observations were recorded for the ground color of testa (GCT), cotyledon color (CC), the color of pattern on testa (CPT), pod dehiscence (PD), pod shedding (PSh), leaflet size (LS), tendril length (TL), ground color of flower (GCF), pod pigmentation (PP), leaf pubescence (LP) and seedling stem pigmentation (SSP). At seedling stage, SSP was observed as '0' for absent and '1' present, LS '3' for small, '5' medium and '7' large, and LP '0' absent, '3' slight and '7' dense. At the early reproductive stage (flowering stage), GCF was recorded as '1' white color, '2' white with blue lines, '3' blue, '4' violet, '5' pink, and '6' for others. At the late reproductive stage, PP was observed as '0' for absent and '1' present. At maturity, PD was categorized as '0' none, '3' low, '5' medium, and '7' high, and PSh '0' none, '3' low, '5' medium and '7' high. Tendril length included '1' for rudimentary tendrils and '2' for prominent tendrils at the late reproductive stage. After harvesting, CPT was observed as '0' absent, '1' olive, '2' grey, '3' brown, '4' black.), CC '1' yellow color, '2' orange-red, and '3' olive green) while GCT represented '1' green, '2' grey, '3' brown, '4' black, and '5' pink.

**Statistical analysis.** Descriptive statistics of quantitative traits including mean, standard deviation, standard error of the mean, range with minimum and maximum values, coefficient of variance, Pearson correlation, and PCA were performed through statistical software XLStat 2019. Hierarchical cluster analysis was conducted to construct dendrograms based on Euclidean distances using RStudio software. The structural equation model (SEM) was drawn through LISREL (LISREL Student Version 9.2 https://lisrel-for-windows-student.software. informer.com/9.2/) to estimate multiple regressions that indicated the contribution of quantitative traits toward yield. Moreover, the genotypic, phenotypic, environmental variances and broad-sense heritability were estimated by the formulas as provided below while genetic advance and genetic advance in the percentage of mean were derived through the variability package in R Studio [28,29].

Genotypic variance

$$GV = \frac{GMS - EMS}{r}$$

Phenotypic variance

$$PV = \frac{GV - EV}{r}$$

Broad-sense heritability

$$h^2 = \frac{GV}{PV}$$

Where GV and PV symbolized the genotypic and phenotypic variances, respectively; GMS, genetic mean square; EMS, error mean square; r, number of replications; EV, environmental variances and $h^2$ represented as broad-sense heritability.

# Result

## Genetic variation based on quantitative traits

High genetic variability was observed among lentil germplasm that could broaden the scope of simple selection and opportunity for new recombinants for genetic improvement. Genotype 6122 collected from Sheikhupura, Pakistan performed the best for SY (68±1.7 g), BY (264±2.8 g), PS (0.61±0.01cm), NSP (2), CT (11 minutes) with no HS. Overall the mean values of SY (30.08±0.98 g) were higher in 2017–18 as compared to 2018–19 (25.26 ±0.87 g) as presented in (Table 1). In comparison, check varieties showed low SY in 2018–19 as compared to 2017–18. It ranged from 5 to 85 g in 2017–18 and 3 to 76 g during 2018–19, respectively. It was noted that genotype 6084 collected from Narowal, Pakistan produced the highest SY in both years while the lowest SY was produced by the genotype 5475 originated from Gujranwala, Pakistan during 2017–18. Likewise, genotype 5494 from Okara, Pakistan, and 5475 from Gujranwala, Pakistan were the lowest in SY during 2018–19. Genotypes 6052 (Syria), 5583 (Muzaffargarh, Pakistan), and 5556 (Hyderabad, Pakistan) produced better SY for both years. The performance of genotypes was highest during the first year as compared to the subsequent year which might be attributed to differences in environmental conditions.

Biological yield is a significant contributor towards SY in lentils and the genotypes with higher biological yield were 6122 (Sheikhupura, Pakistan), 5689 (Sialkot, Pakistan), 5730 (Layyah, Pakistan), and 6037 (Syria) producing 264 g, 258 g, 254 g, and 254 g, respectively during 2017–18, whereas genotypes 6042 (Syria), 6087 (Narowal, Pakistan) and 5689 (Sialkot, Pakistan) were the best with BY of 187.0 g, 183.0 g, and 179 g, in 2018–19. The Plant height

**Table 1. Descriptive statistics for quantitative traits studied in lentil during 2017–18 and 2018–19.**

| Traits | P values | 2017–18 | | | | | | 2018–19 | | | | | |
|---|---|---|---|---|---|---|---|---|---|---|---|---|---|
| | | Mean±SE | Min. | Max. | SD | Markaz 2009 | Punjab 2009 | Mean±SE | Min. | Max | SD | Markaz 2009 | Punjab 2009 |
| SY | 0.0001 s | 25.26±0.8 | 3 | 76 | 12.91 | 39.08±0.9 | 53.27±1.7 | 30.08±0.9 | 5 | 85 | 14.52 | 41.72±1.0 | 55.27±2.4 |
| HSW | 0.254 ns | 1.68±0.01 | 1.17 | 3.28 | 0.28 | 1.85±0.01 | 2.4±0.1 | 1.71±0.02 | 1.05 | 3.65 | 0.35 | 1.91±0.01 | 2.48±0.01 |
| BY | 0.0001 s | 94.13±2.2 | 34 | 187 | 32.77 | 88.58±1.6 | 110.4±5.1 | 114.6±3.0 | 42 | 264 | 44.4 | 87.5±1.8 | 112.2±6.39 |
| PH | 0.0001 s | 46.7±0.4 | 29.6 | 69 | 6.94 | 61.58±1.1 | 56.54±0.9 | 44.73±0.4 | 27.38 | 71.12 | 6.86 | 59.09±1.4 | 55.1±1.4 |
| LPH | 0.958 ns | 14.98±0.1 | 10.72 | 33.34 | 2.03 | 16.56±0.6 | 16.71±0.6 | 14.97±0.1 | 9.14 | 32.16 | 2.34 | 15.32±0.4 | 15.6±0.36 |
| PS | 0.0001 s | 0.3±0.01 | 0.2 | 0.7 | 0.08 | 1.33±0.01 | 1.37±0.01 | 0.3±0.01 | 0.2 | 0.7 | 0.08 | 1.55±0.20 | 1.36±0.01 |
| NSP | 0.0001 s | 2.00±0.01 | 1 | 3 | 0.28 | 2±0.01 | 2±0.01 | 2.00±0.01 | 1 | 3 | 0.16 | 2±0.01 | 2±0.01 |
| DM | 0.0001 s | 186±0.34 | 161 | 198 | 5.05 | 172.5±0.2 | 169±0.3 | 184±0.3 | 161 | 198 | 5.05 | 172±0.29 | 170±0.29 |
| CT | 0.0001 s | 16.15±0.1 | 10 | 18 | 1.52 | 12.16±0.1 | 11.90±1.6 | 14.55±0.1 | 10 | 18 | 1.52 | 12.09±0.1 | 11.7±0.19 |
| HS | 0.0001 s | 0.41±0.04 | 0 | 2 | 0.61 | 0.41±0.1 | 0.09±0.9 | 0.37±0.01 | 0 | 2 | 0.61 | 0.27±0.1 | 0.25±0.17 |
| NP | 0.0001 s | 2.00±0.01 | 1 | 3 | 0.16 | 3±0.01 | 3±0.01 | 2.04±0.01 | 2 | 3 | 0.19 | 3±0.01 | 3±0.01 |

SY, seed yield; HSW, hundred seed weight; BY, biological yield; PH, plant height; LPH, lower pod height; PS, pod size; NSP, number of seeds per pod; DM, days to maturity; CT, cooking time; HS, hard seed/hundred seed; NPP, number of pods per peduncle; SE, standard error of mean; SD, standard deviation; p-value <0.05.

(PH) was slightly lower during 2018–19 as compared to 2017–18, PH has a significant contribution towards acquiring light and the high number of branches to produce more seeds.

Days to maturity ranged from 161 to 198 days in both years, however, mean values during 2017–18 were 184 days whereas 186 days during 2018–19. The genotypes were 5684 (Narowal, Pakistan), 5988 (USA), 5979 (USA), 5679 (Narowal, Pakistan), 5982 (Narowal, Pakistan), 5683 (Narowal, Pakistan), 24786 (unknown), 5623 (Faisalabad), 6077 (Bahawalnagar, Pakistan) and 5684 (Narowal, Pakistan). These genotypes can be considered for fodder as they meet the requirement for longer vegetative growth. Short duration genotypes including 5698, 6015, 5667, 5748, 5555, 6017, 5700, 5571, 5580 and 5981 matured in < 178 days.

Both the check varieties were insignificant within the same year, however, overall performance was better during 2017–18 as compared to 2018–19. It indicated homogeneity of the experimental site and the validity of augmented design due to higher genetic variance. Superior genotypes for individual traits have been identified and presented in Table 2, which are

**Table 2. List of best performing lentil genotypes selected during both years for quantitative traits.**

| Traits | During 2018 | During 2019 |
|---|---|---|
| SY > 40g | 6084, 6062, 6122, 6058, 6087, 5689, 6042, 6074, 5664, 5687 | 6084, 6062, 6122, 6058, 6087, 5689, 6074, 6042, 5664,5687, |
| HSW > 1.90g | 6052, 5583, 5556, 6101, 5684, 23776, 5643 | 5583, 6052, 5684, 6101, 5556, 5643, 23776 |
| BY > 100g | 6122, 5689, 6037, 6042, 5600, 5982, 5658, 6084 | 6122, 5689, 6037, 6042, 5600, 5982, 5658, 6084 |
| PH > 55 | 5549, 5687, 5723, 5549, 5687 | 5549, 5687, 5723, 5549, 5687 |
| LPH > 15cm | 5527, 6062, 5751, 5686, 5744, 5477 | 5527, 6062, 5751, 5686, 5744, 5477 |
| PS > 0.4 cm | 5664, 6124, 6116,6077, 6082 | 5664, 6124, 6116,6077, 6082 |
| NSP > 2 | 6122, 5687, 24784, 6012, 6084 | 6122, 5687, 24784, 6012, 6084 |
| DM < 180 days | 24783,5741, 5729, 6081, 5981 | 24783, 5741, 5729, 6081, 5981 |
| CT < 12 minutes | 6074,6075,6010, 6013, 6041 | 6074,6075,6010, 6013, 6041 |
| HS < 2 | 6074, 5595, 6075, 6010,6041 | 6074, 5595, 6075, 6010,6041 |
| NP > 2 | 5595,5861, 5691, 5671,5689 | 5595,5861, 5691, 5671,5689, |

SY, seed yield; HSW, hundred seed weight; BY, biological yield; PH, plant height; LPH, lower pod height; PS, pod size; NSP, number of seeds per pod; DM, days to maturity; CT, cooking time; HS, hard seed/hundred seed; NPP, number of pods per peduncle.

**Table 3. Frequency distribution of qualitative traits of lentil germplasm studied during 2017–18 and 2018–19.**

| Variable | No. of Categories | Categories Name | Frequency (%) | Variable | No. of Categories | Categories Name | Frequency (%) |
|---|---|---|---|---|---|---|---|
| Ground color of testa (GCT) | 4 | Green | 1.4 | Pod shedding (PS) | 3 | Low | 55.9 |
| | | Grey | 51.8 | | | Medium | 26.8 |
| | | Brown | 33.6 | | | High | 17.3 |
| | | Black | 13.2 | Leaflet size (LS) | 3 | Small | 62.7 |
| Cotyledon color (CC) | 3 | Yellow | 2.3 | | | Medium | 28.2 |
| | | Red | 96.4 | | | Large | 9.1 |
| | | Olive green | 1.4 | Pod pigmentation (PP) | 2 | Absent | 69.1 |
| Color of pattern on testa (CPT) | 5 | Absent | 0.9 | | | Present | 30.9 |
| | | Olive | 7.7 | Leaf pubescence (LP) | 2 | Slight | 85.0 |
| | | Grey | 59.5 | | | Dense | 15.0 |
| | | Brown | 13.6 | Seedling stem pigmentation.(SSP) | 2 | Absent | 69.1 |
| | | Black | 18.2 | | | Present | 30.9 |
| Pod dehiscence (PD) | 4 | None | 0.9 | | | | |
| | | low | 87.7 | | | | |
| | | Medium | 8.2 | | | | |
| | | High | 3.2 | | | | |

recommended for future evaluation under wide range environments for selection of superior cultivar/s. More than 40 g SY was produced by three genotypes 6084, 6062, and 6122 during both years, hence selected for utilization of yield potential. Cooking time is very crucial characteristics for grains, particularly in the era of energy crises and higher fuel prices, hence short CT is an emerging selection criterion for most legumes including lentils. Seven genotypes *viz.*, 6074, 5595, 6075, 5518, 6092, 6099, and 5511 which took <12 minutes for cooking were shortlisted for future utilization. A list of genotypes representing the best genotypes based on five percent selection criterion including check varieties is presented (S2 and S3 Tables).

**Genetic variation in qualitative traits.** High variation was observed for qualitative traits as various lentil plant descriptors were described by dividing them into categories (Table 3). The ground color of testa (GCT) exhibited four categories, out of which more than half of lentil germplasm have grey color, 33.6% as brown, 13.2% black while few genotypes (1.4%) were green. Cotyledon color (CC) was predominantly red, consumers prefer red cotyledon color which has more market demand.

**Pearson correlation matrix.** The correlation matrix for quantitative traits presented in Table 4 revealed a positive association of SY with BY, PS, NSP, and NP. Significant contribution to SY by importat traits indicated the effectiveness of yield contributing traits that can be translated to boost yield and for selection of genotypes with desirable traits. In the same context, lentil genotypes grown in 2018–19 indicated that SY exhibited a significant positive correlation with HSW, BY, and PH and rest assure that these traits contributed directly towards seed yield.

## Principal component analysis (PCA)

First five components contributed > 1.0 eigenvalue as 2.09, 1.37, 1.19, 1.08 and 1.02 respectively based on the PCA for 2018 (S4 Table). The SY, BY and PS contributed more toward PC 1, whereas component 2 contributed more for HSW, PH, and HS. The PCA facilitates further selection of genotypes with specific trait/s without losing genetic variation. Based on second year PCA, factor 1 was more contributed by SY (0.84), BY (0.84), and NP (0.02), hence the

**Table 4. Pearson Correlation matrix for SY with other quantitative traits of lentil grown in 2017–18 and 2018–19.**

| Traits | Years | HSW | BY | PH | LPH | PS | NSP | DM | HS | NP |
|--------|-------|-----|-----|-----|------|-----|------|-----|------|-----|
| SY | 2018 | 0.09 | 0.73* | -0.09 | 0.03 | 0.32* | 0.15* | 0.15* | 0.11 | 0.18* |
| | 2019 | 0.27* | 0.83* | 0.21* | -0.16* | -0.11 | -0.11 | -0.05 | -0.01 | 0.08 |

SY, seed yield; HSW, hundred seed weight; BY, biological yield; PH, plant height; LPH, lower pod height; PS, pod size; NSP, number of seeds per pod; DM, days to maturity; HS, hard seed; NP, number of pods per peduncle.

genotypes falling in PC 1 are affected by reproductive traits rather than vegetative traits, while factor 2 is contributed through HSW (0.32), PS (0.63) and DM (0.15) as shown in Fig 1.

Furthermore, scatter plot based on the first two factors presented in Fig 2 showed that the genotypes *viz.*, 5638, 5583, 6124, 6052, 6010, 5684, 5979, and 6101 were present in the extreme

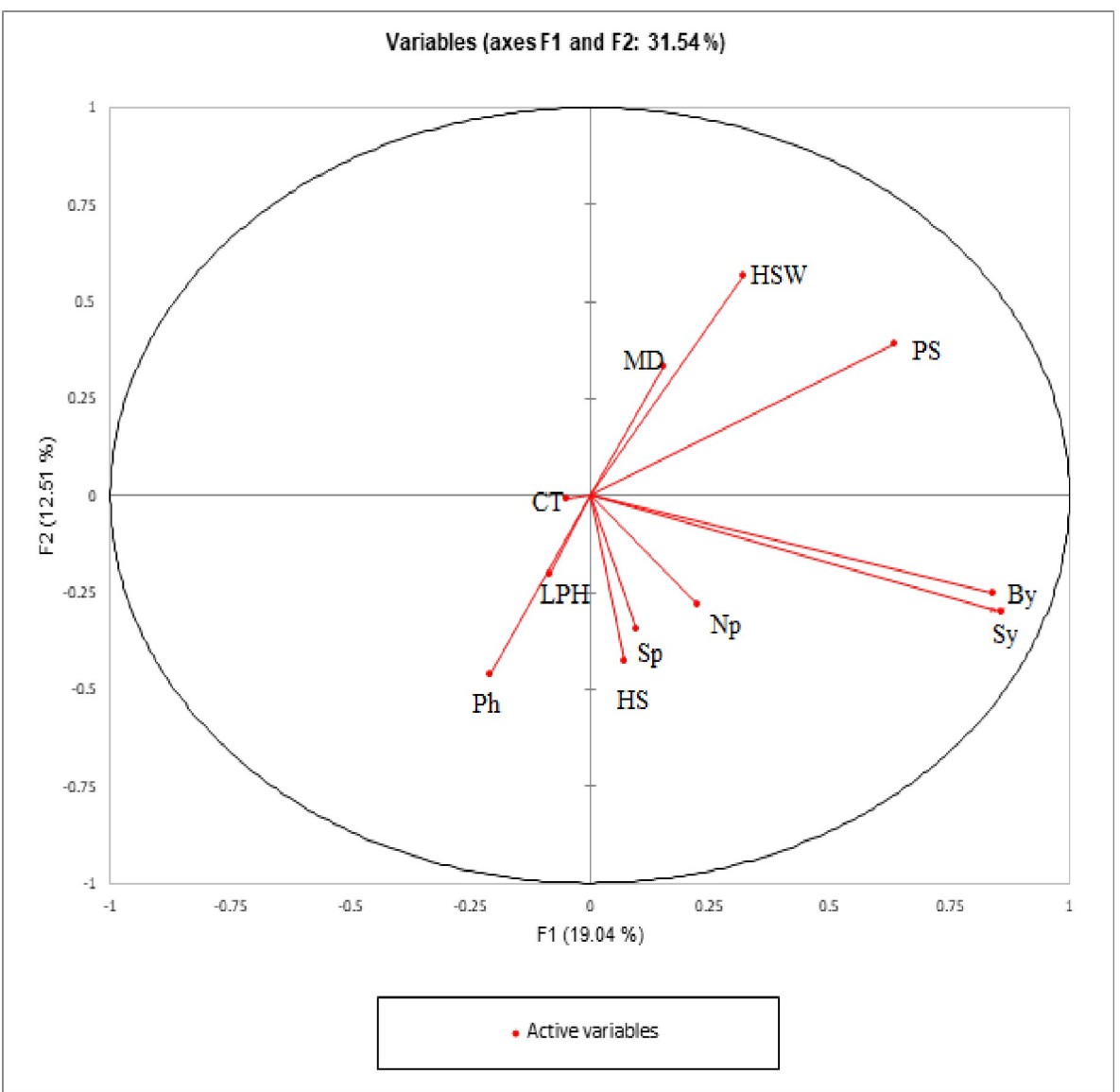

**Fig 1. PCA showing the contribution of quantitative traits among 220 lentil genotypes grown in 2017–18.**

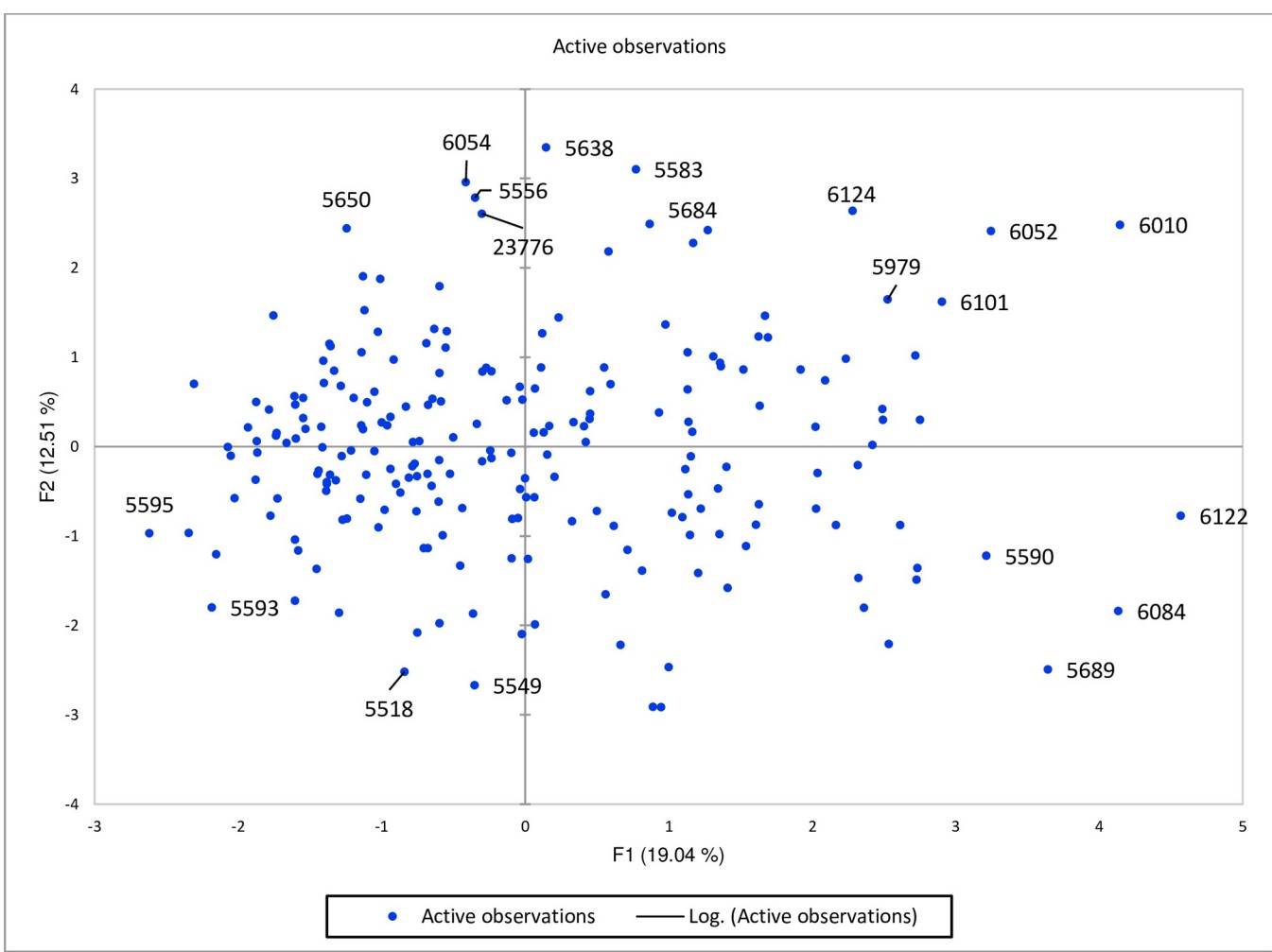

**Fig 2. A scattered plot of PCA for 220 lentil genotypes studied during 2017–18.**

upper right side of the scattered plot due to response towards HSW, DM, and PS. The genotypes 6054, 5556, 23776, and 5650 were grouped at an extreme upper left side with a mixed performance for various traits. These genotypes are required further evaluation and characterization to explore the full potential. Similarly, the genotypes 6122, 6084, 5689, and 5590 on the lower right side of the scattered plot were more contributed through BY, SY, NP, NSP, and HS.

Moreover, based on PCA during 2019, the first five components had > 1 Eigenvalue, whereas other PCs contributed low in cumulative variation (S5 Table). Contribution by SY, BY, and HSW were the highest toward factor 1 while factor 2 contributed more by CT, SP, and HS. The separation of genotypes and their accumulation in the particular principal components indicated the performance and association towards yield. Resultantly, factor 1 showed 19.91% variability while factor 2 as 12.19% with cumulative contribution of 32.10% (Fig 3).

The genotypes 5506, 5729, 5510, 5672, 5636, 5581, 5491, and 5483 at the upper right corner were grouped based on HSW, BY and SY indicating the scope of selection for elite genotypes on these genotypes, while in the lower right corner genotypes 5472, 5512, 5479, 5486, 5650, 5600 and 5531 indicated the importance of PH, NP, and HS for selection. However, resultantly, an increase in plant height would icrease seeds per pod that ultimately contributes towards seed yield. At the upper left side of the scattered plot, genotypes 6080, 6043, 5653,

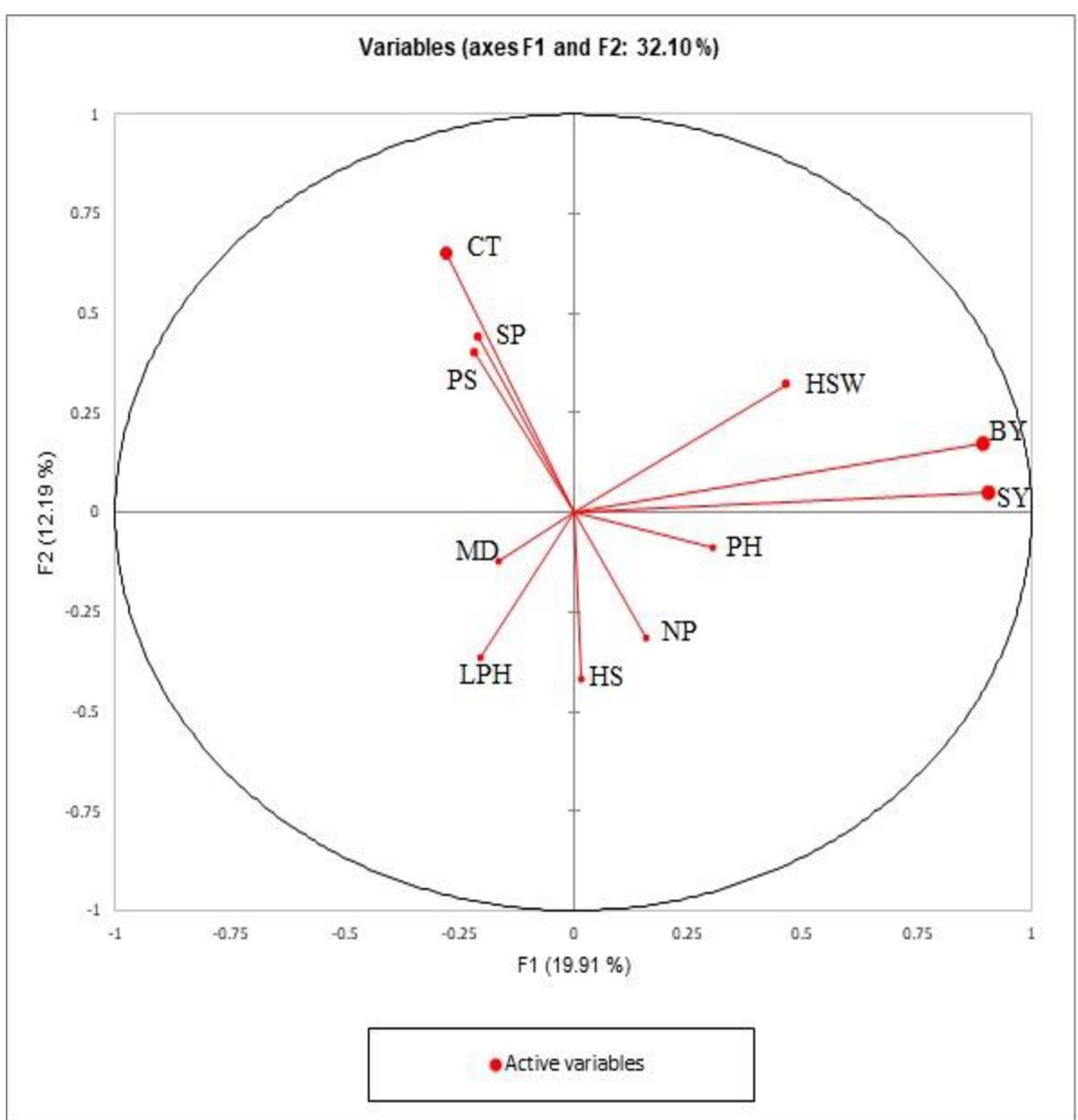

**Fig 3. PCA showing the contribution of quantitative traits among 220 lentil genotypes grown in 2018–19.**

6043, 6002, 6060, and 5677 were grouped which were contributed through CT, NSP, and PS, hence these could be selected for appropriate CT to save energy for cooking lentil. On the other hand, genotypes 23787, 24787, 23779, 5742, and 5712 were grouped in the lower left corner exhibiting higher variability for DM and LPH which supports to select genotypes for these traits (Fig 4). The lower pod height is an important trait particularly for developing short duration lentil cultivars due to stature of the plant that helps to penetrate more light for high metabolism.

## Hierarchical cluster analysis

Cluster analysis based on quantitative data in 2017–18 is presented in Fig 5 which had divided the genotypes into 4 main clusters based at 50% linkage distance (LD) and cluster 1 consisted

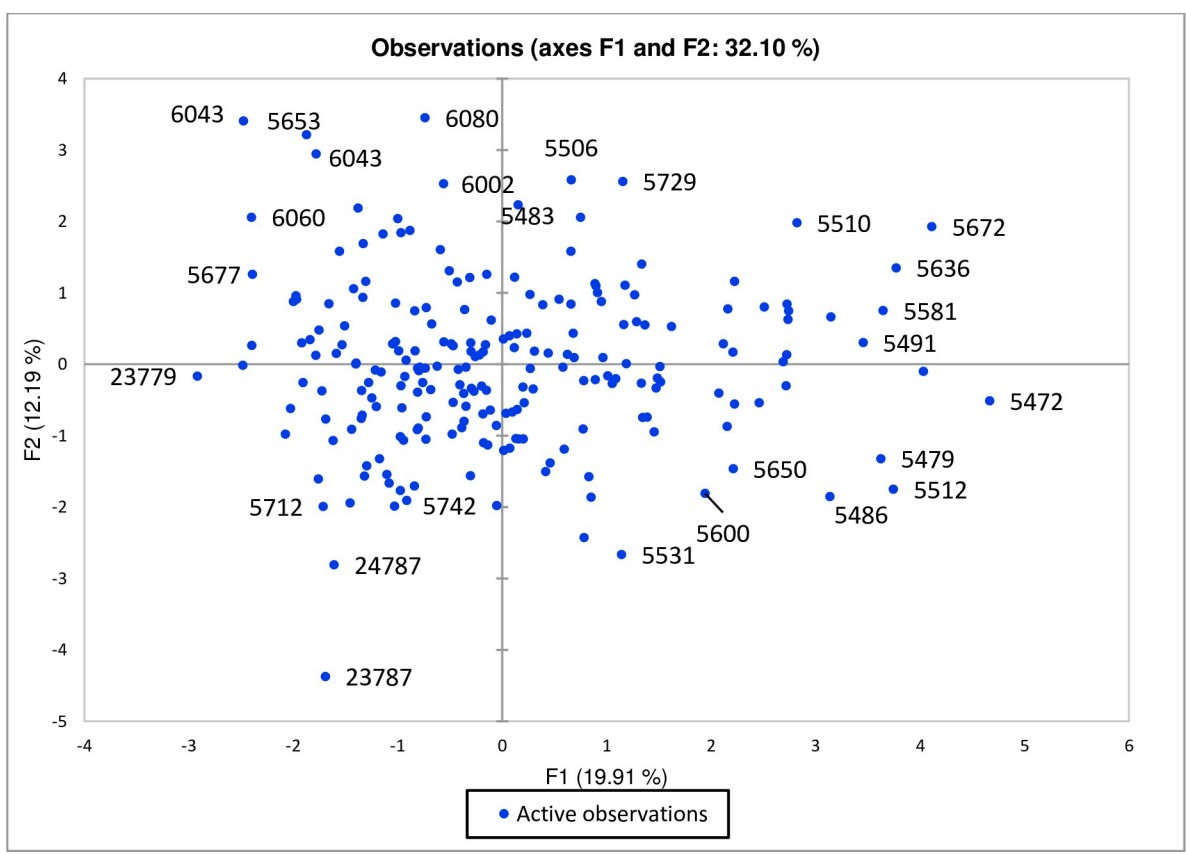

**Fig 4. A scatter plot of PCA for lentil genotypes grown in 2018–19.**

of 106 diverse genotypes including 6015 originated from the USA followed by cluster 2 contains 85 genotypes including 5698 originated from Sialkot, Pakistan with a distinct position. Cluster 3 consisted of 20 genotypes where 3 representing Pakistan were found to be distinct (5689 Sialkot, 5687, and 6084 from Narowal). Nine genotypes with unique performance for SY, BY, PS, NSP, CT, and HS were grouped together in cluster 4 and these genotypes have been selected for their unique performance, hence suggested to be used for future lentil breeding.

The hierarchical cluster pattern for 2018–19 indicated 7 main clusters at a 50% linkage distance (Fig 6). Cluster 1 consisted of 37 genotypes, cluster 2 (115), cluster 3 (12), while clusters 4 and 5 were of 1 (24783) and 4 genotypes (6084, 6058, 6062, and 6122), respectively. The cluster 6 consisted of 49 genotypes, and cluster 7 had only 2 genotypes. Best performing genotypes were grouped in C3 during 2017–18 and C5 during 2018–19 and seven traits (SY, BY, PS, NSP, CT, and HS) contributed toward genotypic performance. The cluster 2 consisted of those genotypes which had PH > 55.1, DM > 170 and HSW > 1.18 during both the years, however, cluster 1 consisted of high yielding genotypes having SY > 40, hence these geotypes were identified for future testing under wide range of environmental conditions.

## Structural equational model (SEM)

The SEM constructed based on covariance keeping SY as dependent variable and HSW (P values 0.10), BY (0.09), PH (0.00), LPH (0.01), PS (0.00), DM (0.00), NP (0.04), and NSP (0.81) as independent variables indicated varying degrees of contribution by independent variables

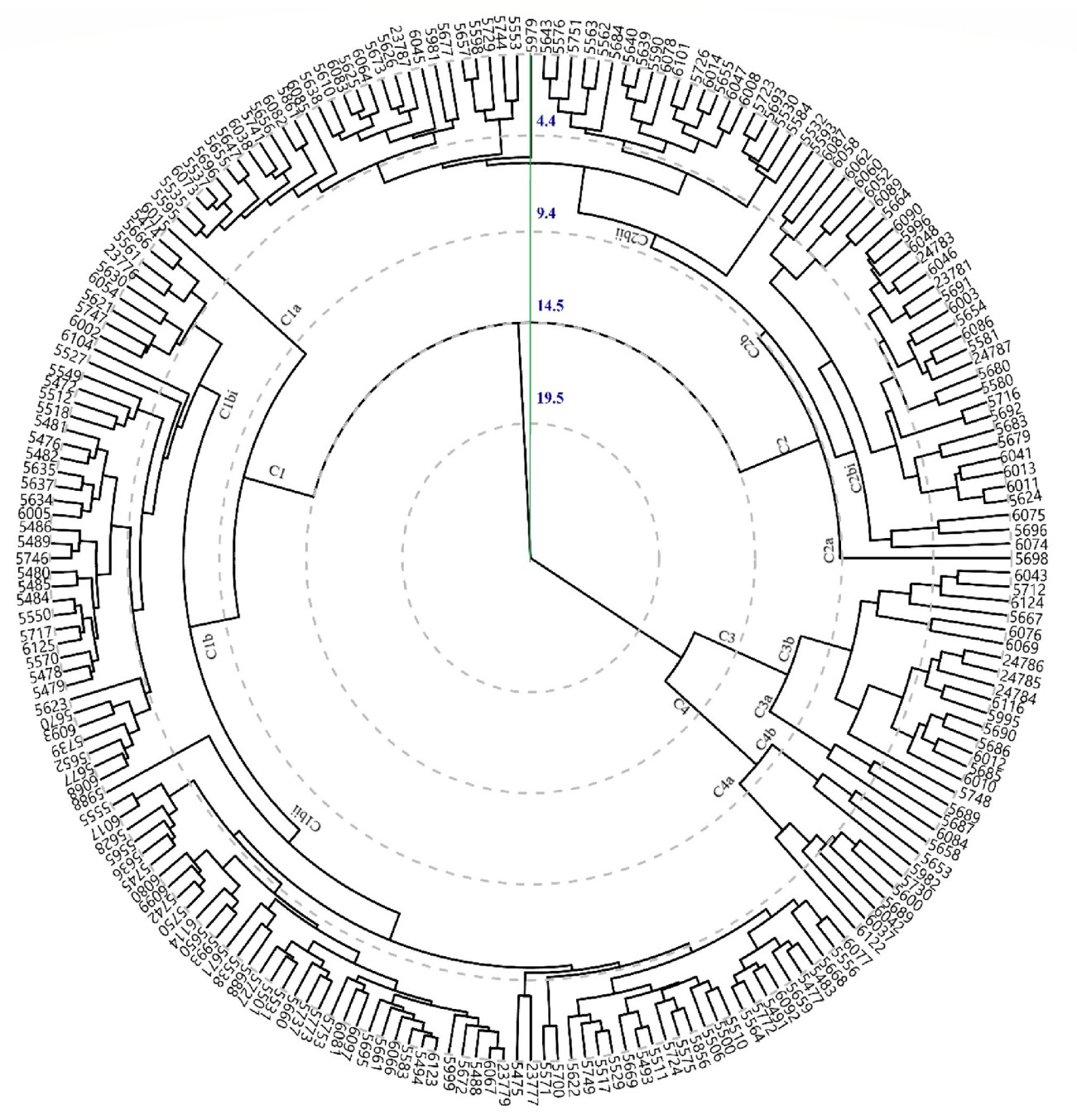

**Fig 5. Hierarchical dendrogram showing the relationship of lentil genotypes for quantitative traits studied during 2017–18.**

(Table 5). These results were in coordination when verified by the lowest error variance of multiple linear regressions. Coefficient of determination ($R^2$ = 0.64) for multiple linear relationships revealed that real yield contributors as given in the equation contributed in percent capacity share, as shown in the Fig 7.

**Heritability analysis.** Higher magnitude of phenotypic coefficient of variance for all the traits during both the years indicated the environmental effects on the experiments under study (Table 6). High heritability was observed for most of the traits, however Number of

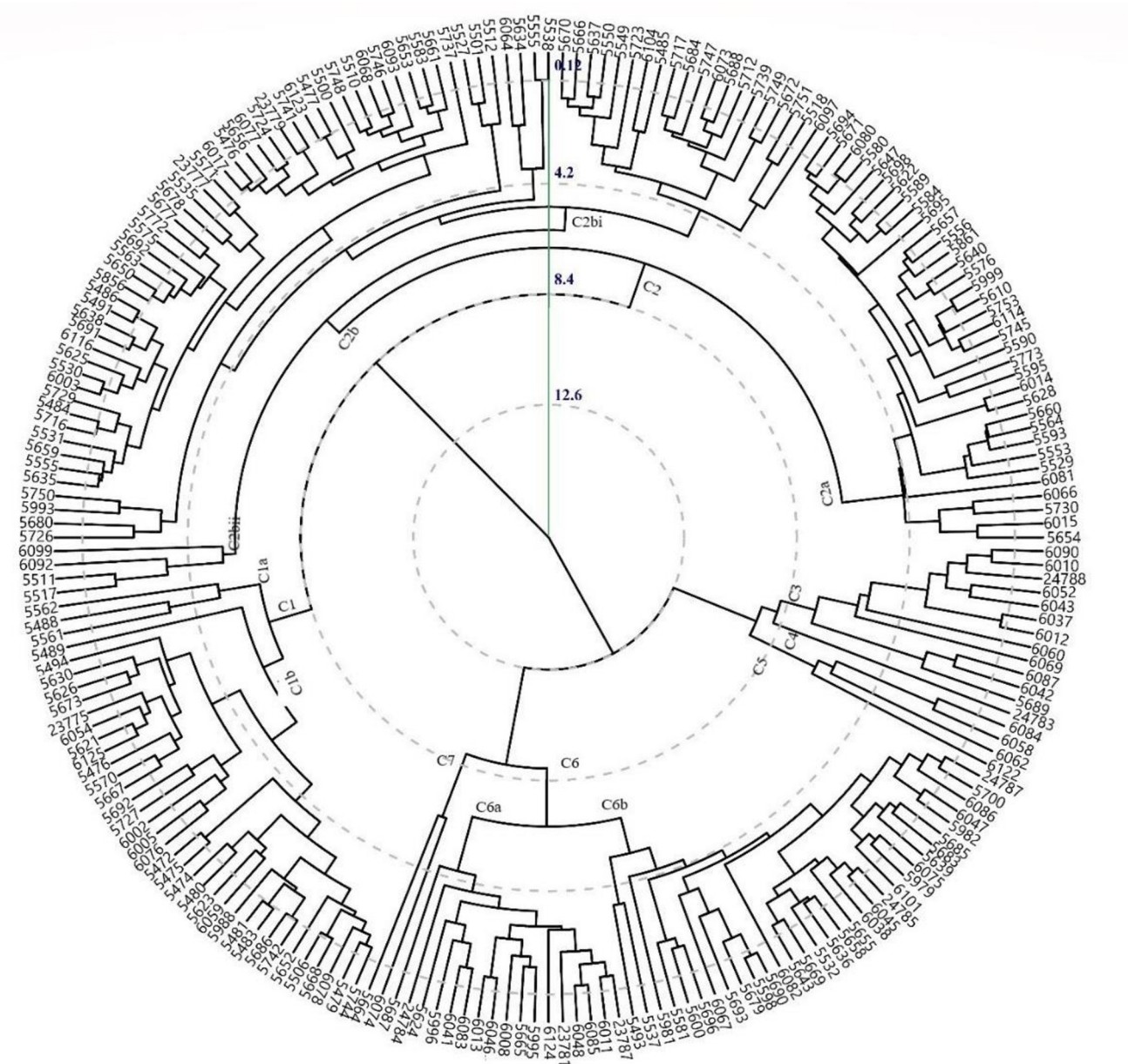

**Fig 6. Hierarchical dendrogram showing the relationship of lentil genotypes for quantitative traits studied during 2018–19.**

**Table 5. Structural equational model representing regression for agronomic traits studied during 2017–18 and 2018–19.**

| SY | HSW | BY | PH | LPH | NSP | PS | DM | NP |
|---|---|---|---|---|---|---|---|---|
| Z-values | 2.12 | 0.27 | 0.02 | 0.43 | 3.69 | 5.43 | -0.04 | 2.21 |
| P-values | 0.10 | 0.09 | 0.00 | 0.01 | 0.81 | 0.0 | 0.0 | 0.04 |

SY = 2.16*HSW + 0.27*BY + 0.025*PH + 0.0.43*LPH + 1.826*NSP + 6.34*PS—0.039*DM + 2.298*NP, SY, seed yield; BY, biological yield; PH, plant height; LPH, lower pod height; PS, pod size; DM, days to maturity; NP, number of pods per peduncle; NSP, number of seeds per pod.

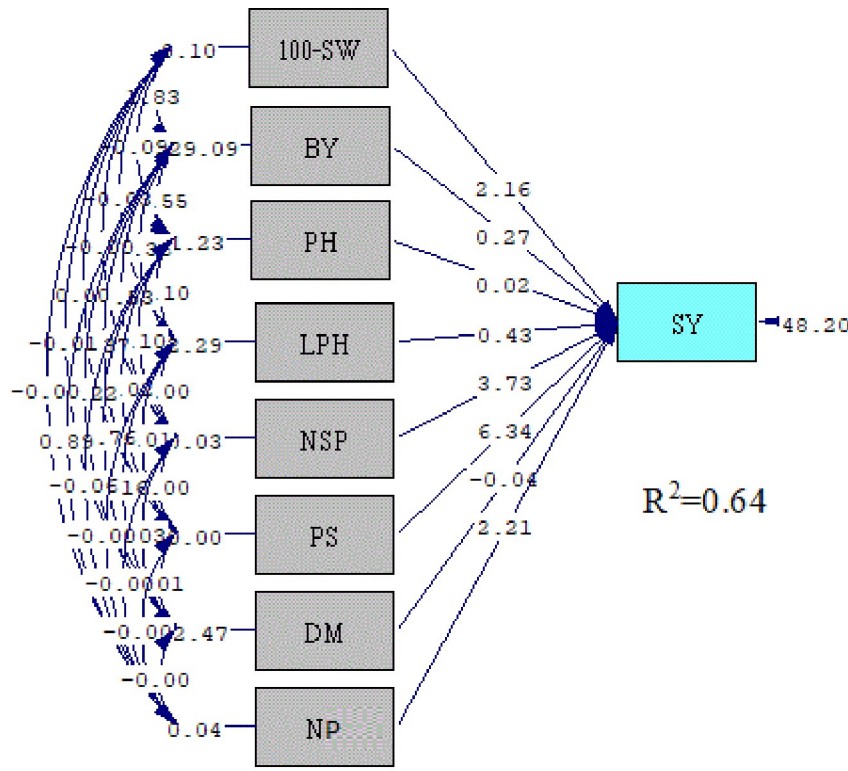

Chi-Square=0.00, df=0, P-value=1.00000, RMSEA=0.000

**Fig 7. Structural equations model for seed yield (SY) during 2017–18 and 2018–19.**

seeds per pod (NSP) showed a low heritability value of 0.46 during 2017–18 as compared with 2018–19 at 0.88 that revealed the impact of environmental fluctuations for development of NSP. Other traits showed high heritability such as 0.99 for BY, PH, LPH, and DM during 2017–18 while during 2018–19 LPH and DM showed the same heritability value. The high

**Table 6. Estimation of heritability and genetic advance for the years 2017–18 and 2018–19.**

| Traits/Years | Environmental Variance | | Genotypic Variance | | Phenotypic Variance $(\sigma_p)$ | | Environmental Coefficient of Variance | | Genotypic Coefficient of Variance | | Phenotypic Coefficient of Variance | | Heritability (Broad Sense) $(h^2)$ | | Genetic Advance | | Genetic Advance as a percentage of mean | |
|---|---|---|---|---|---|---|---|---|---|---|---|---|---|---|---|---|---|---|
| | 2018 | 2019 | 2018 | 2019 | 2018 | 2019 | 2018 | 2019 | 2018 | 2019 | 2018 | 2019 | 2018 | 2019 | 2018 | 2019 | 2018 | 2019 |
| SY | 6.21 | 3.75 | 206 | 163.9 | 212.9 | 167.72 | 8.23 | 7.69 | 47.47 | 50.84 | 48.18 | 51.41 | 0.97 | 0.97 | 29.18 | 26.08 | 96.36 | 103.55 |
| HSW | 0.01 | 0.00 | 0.11 | 0.07 | 0.12 | 0.08 | 7.64 | 5.66 | 19.3 | 15.83 | 20.80 | 16.81 | 0.86 | 0.88 | 0.63 | 0.51 | 37.06 | 30.71 |
| BY | 16.9 | 33.09 | 1978 | 1050.9 | 1995 | 1084. | 3.59 | 6.11 | 38.85 | 34.43 | 39.02 | 34.97 | 0.99 | 0.96 | 91.24 | 65.75 | 79.71 | 69.85 |
| PH | 0.46 | 0.61 | 47.04 | 48.10 | 47.5 | 48.72 | 1.51 | 1.61 | 15.26 | 14.23 | 15.34 | 14.32 | 0.99 | 0.98 | 14.06 | 14.19 | 31.29 | 29.13 |
| LPH | 0.02 | 0.02 | 5.52 | 4.16 | 5.54 | 4.18 | 1.01 | 1.10 | 15.87 | 13.61 | 15.90 | 13.65 | 0.99 | 0.99 | 4.83 | 4.18 | 32.63 | 27.94 |
| PS | 0.00 | 0.00 | 0.01 | 0.01 | 0.01 | 0.01 | 3.07 | 4.88 | 20.4 | 21.92 | 20.70 | 21.92 | 0.97 | 0.95 | 0.17 | 0.16 | 41.48 | 44.24 |
| NSP | 0.01 | 0.00 | 0.01 | 0.07 | 0.02 | 0.08 | 5.69 | 3.21 | 5.34 | 13.66 | 7.81 | 14.04 | 0.46 | 0.94 | 0.15 | 0.56 | 7.52 | 27.40 |
| DM | 0.13 | 0.13 | 25.60 | 25.60 | 25.73 | 25.74 | 0.19 | 0.19 | 2.73 | 2.73 | 2.74 | 2.74 | 0.99 | 0.99 | 10.39 | 10.39 | 5.62 | 5.62 |
| HS | 0.00 | 0.01 | 0.37 | 0.36 | 0.38 | 0.38 | 25.88 | 30.52 | 163.9 | 163.12 | 165.9 | 165.9 | 0.97 | 0.96 | 1.24 | 1.23 | 333.5 | 330.32 |
| NPP | 0.01 | 0.00 | 0.04 | 0.02 | 0.05 | 0.03 | 5.50 | 2.51 | 9.69 | 7.90 | 11.15 | 8.29 | 0.75 | 0.90 | 0.35 | 0.31 | 17.36 | 15.53 |

SY, seed yield; HSW, hundred seed weight; BY, biological yield; PH, plant height; LPH, lower pod height; PS, pod size; NSP, number of seeds per pod; DM, days to maturity; HS, hard seed; NPP, number of pods per peduncle.

heritability was observed for BY (0.99) along with high genetic advance as a percentage of the mean of 79.71 for 2017–18 while for the subsequent year LPH was 27.94. Higher heritabilities for most of the traits indicates the predominance of additive gene action for controlling these traits, hence simple selection from this diversified material could be successfully employed for lentil improvement. High genetic advance along high heritability estimates provides ample opportunity for suitable selection of traits for future application, either through simple selection of parental lines to be used in hybridization for varietal development.

## Discussion

Selection of promising genotypes is the ultimate objective of plant breeders to develop high yielding cultivars for farmers' fields and ultimately for enhancing crop productivity. Genetic diversity has been used to characterize crop germplasm in a range of plant species including lentils [30]. High genetic diversity was observed for SY ranging from 3 to 85 g and on the basis of two years' performance, the genotype 6084 was identified as the best yielding among lentil germplasm and selection on the basis of evaluation has been previously reported [16]. It is always desired by breeders to choose high yielding genotypes at initial evaluation stage and then could be tested under multi-location trials for varying environments [31]. In most of legumes, the BY is not in linear correlation with SY, however few genotypes may have positive association between these important traits, that induces increase of photosynthetic area to convert into SY [32]. In the present study, 6122, 5689, 5730, and 6037 exhibited the best BY during 2017–18, and genotypes 6042, 6087, and 5689 were best during 2018–19. Higher BY at vegetative growth stage before flowering leads to further increase in N-assimilation and thereby enhance SY [24,33,34]. Nitrogen assimilation transfer atmospheric nitrogen into amino acids, therefore it is one of the natural property of legumes that make them more proteinaceous food [35].

Overall performance of lines was better in the first year as compared to the subsequent year that is more likely attributed to temperature fluctuation during both of the years, from 21˚C in March to 26˚C in April during 2017–18 and 19˚C in the same duration. It is obvious that low-temperature delay the flowering and pod filling process, and lentil requires low temperature during the vegetative stage, while high temperatures at reproductive stage [36]. The optimum temperature for the vegetative stage of lentils ranged from 6.1–27.3˚C while 17.4–31.5˚C during the reproductive stage [37]. Likewise, plant height (PH) is a major yield contributing trait which supports sufficient vegetative growth and branching and ultimately leading to development of more pods which means increased SY [7,38].

High variation during both years were additionally influenced due to environmental factors including water scarcity, drought during reproductive and grain filling stages that resultantly decreased SY during second year along with negative effects on yield contributing traits [39,40]. In this study, genotypes 5698 (Sialkot, Pakistan) and 6015 (USA) were early maturing in 2017–18 while 24783 (unknown) and 5561 (Sanghar, Pakistan) for 2018–19 with 161–172 days. However, current germplasm was 10–20 days early in maturity as previous reports of 168–190 days [16]. The early maturing genotypes are suggested for crop improvement programs in drought prone areas [41].

It is evident that SY is directly proportional to NSP, NP ranged in both the years, hence selection based on NP would be the baseline material for multi-location trials under different environmental conditions to develop new varieties [17]. Similarly, HSW is considered an indicator of seed quality and a measure of grain density. In the present study the average HSW varied 3.65 g in 2017–18 and 3.28 g in 2018–19, hence bold seeded lentil cultivars can be selected from the germplasm evaluated for two years. Although small to medium seed grains are

traditionally preferred by local communities, however due to imports of bold seeded lentil from Turkey, Australia and Canada, now bold seeded lentil has huge market in Pakistan, hence the germplasm under present studies has many bold-seeded genotypes with better yield potential which have been identified for future use [3,7,11,14–18,39,42].

Furthermore, cooking time (CT) is a favorite trait for consumers in the case of legume crops because it saves energy. CT of < 12 minutes was observed in genotypes 6074, 6075, 6010, 6013, and 6041. It will not only saves energy and time but also will develop adequate flavor, texture and improve protein digestibility in lentils [8]. Higher CT causes loss of proteins, vitamins, minerals, and other secondary metabolites [9]. Conclusively, the genotype 6122 performed the best among all the germplasm therefore, this is suggested for future lentil breeding programs to develop new variety for commercial purposes.

Inheritance studies of qualitative traits are like the plant descriptor and are more influenced by natural selection and consumers' preference, as in lentil GCT and GPT which are controlled by two independent alleles [43]. The dominant allele at one locus (Ggc) determines grey seed coat color and the dominant allele at the second locus (Tgc) determines tan seed coat color. Two dominant alleles (Ggc Tgc) control brown GCT and double recessive (ggc tgc) has green color. The seed coat pattern is highly complex qualitative trait determined by five alleles at various loci [41]. Moreover, pod pigmentation was absent in 69% of genotypes with low PD and PSh. Low PD and PSh are the most favorite traits from farmer's perspective because they reduce grain loss during harvesting. Further, slight LP with a small leaf and prominent tendril as was most frequently found. While dense LP (leaf pubescence) was in 15% genotypes that reduces the absorbance of photosynthetically active radiation (400 to 700 nm), hence non-hairy or slight LP is an effective trait for photosynthetic metabolism and light absorption rather than decreased carbon dioxide conductance through the boundary layer [17]. It has been reported that slight LP with small leaves with prominent tendrils exhibit higher resistance to lodging. Seedling pigmentation was present in most of the genotypes and the genotypes exhibited white blue line flowers as already reported [42].

Improvement in a target character can be achieved by indirect selection that is highly heritable and easier to select, however, this strategy of selection requires understanding of interrelationships among characters [43]. Significant correlation of SY with most of the quantitative traits was documented in both years and positive combinations can be helpful in future hybridization programs to obtain better recombinants. It is believed by many researchers that seed yield is mainly associated with increase in seed germination, NP, and seed index, and in contrary, prolonged flowering and delayed maturity in lentil crop reduces SY [44,45]. Therefore, it is important to plant lentil within the optimum window. The number of pods per peduncle had a significant positive correlation with SY that is needed to be exploited as suggested [16,46].

Based on the PCA traits, *viz*., SY, BY and PH contributed maximum toward Factor 1 that resulted in grouping of genotypes mainly on reproductive traits, whereas factor 2 was mainly contributed by vegetative traits and seed traits, recommended for further evaluation under various locations for selection of the best genotype. Similar results were observed in lentils collected from Pakistan and Turkey in the past [16,17]. Scattered diagrams reflected high variation and genotypes with better performance for SY, BY, PS, HSW, PH, and HS were grouped together and more that half of cumulative variance was contributed by first two principal components [24,47,48]. It was revealed that Factor 11 indicated a high potential for both SY and BY, therefore this component is also suggested for further use as lentils with desirable yield and mass for fodder uses. Factor 4 more contributed for maturity and other factors exhibited negligible contribution for various traits. Genotypes with best CT placed in factor 6 during 2017–18 and factor 2 during 2018–19. The correlation matrix for DM showed a

negative association with CT and is supported by the fact that time of maturity is always influenced by variation in environmental factors. Hence selection for genotypes with the best CT from different components during both years can be accomplished at random. Individual components can also be selected for NP, LPH, and PH which were placed in factors 3, 4, and 5, respectively. It can be derived that PCs which showed consistent performance in both years would be more beneficial to cater genotypes potential.

Hierarchical cluster analysis revealed that genotypes, *viz.*, 6015 (USA) and 5698 (Sialkot, Pakistan) showed an outlier from all other clusters in 2017–18 that was more influenced by environmental effects for DM. Genotypes 5664 (Sialkot, Pakistan) and 6074 (Bahawalnagar, Pakistan) were distinct from others in 2018–19, resultantly placed in cluster 7. These genotypes were the best for most of the traits including SY. It was observed that Syrian germplasm was grouped into cluster 2 during 2018–19 while collections from USA and Pakistan were grouped into different clusters. Grouping of genotypes from various origins might be due to frequent germplasm exchange through ICARDA or might be due to common ancestry. It is assumed that grouping of genotypes during both the years in separate clusters was due to environmental conditions especially temperature at the experimental site, however, promosing genotypes that carries genes of interest are suggested to be exploited for lentil breeding program [49–51]. Nevertheless, diverse genotypes are suggested to produce transgressive segregates keeping in consideration inter-cluster distance between that would be more genetically divergent [52,53].

Structural equation model illustrates association among multifactorial traits which drive that SY is dependent on other traits in a casual and effective way explaining path of the characters towards SY. In the current findings, the SEM showed a significant level $< 1$ contribution of independent variables like LPH, PS, DM, HS, NP, and NSP toward SY which can exploited through simultaneous selection on the basis of independent variable for cumulative effect of dependent variable [16]. Higher levels of phenotypic coefficient of variance was more likely be attributed due to environmental factors, however, most of the agro-morphological traits had fairly higher heritability indicating additive genetic variance to exploit through selection [18]. Traits with high heritability indicate potential for utilizing additive genetic variation and traits with low heritability need to be improved through bi-parental selective mating using heterotic breeding [54,55], however, for the traits (SY, BY and LPH) with high heritability coupled with high genetic advance are suggested to be exploited through simple selection for crop improvement [56]. Phenotypic variance inferred influence of environmental factors for heritable quantitative traits, hence, traits with high heritability are more important for starting varietal development program [57].

## Conclusion

Genotype 6122 originated from Sheikhupura, Pakistan was the best for most of the traits during both the years, hence recommended for general cultivation after completion of formal requirements and experimentations. Yield per plant was recorded as 8.05 g and ultimately provides estimates of yield tons/hectare as 2.68, while check variety Markaz-09 has a yield potential of 3.2 tons/hectare, advocates high future prospective for current lentil germplasm. Genotypes representing Pakistan and Syria exhibited the best values for BY which possess the potential to be used as fodder or forage purposes, particularly during cooler months of fodder scarcity. The present investigation reported DM range of 161–198 days during both of the years, however earlier reports showed more time for maturity, hence short duration genotypes (5698, 5667, 5748, 5555, 5700, 5571, 5580, 5981, 6015, 6017,) are recommended for further testing for selecting early maturing lentil cultivars. Keeping in view, high genotypic coefficient of variance (GCV) along with high heritability and genetic advance depicted wide variation for

selection with improving yield traits in lentil cultivars. Ten genotypes *viz.*, 5664, 5687, 5689, 6042, 6058, 6062, 6074, 6084, 6087, and 6122, were selected based on higher potential for yield, hence recommended for future crop improvement programs.

## Supporting information

**S1 Fig. Average temperature during lentil crop season October to April 2017–18 and 2018–19 at the experimental station.**
(DOCX)

**S1 Table. List of lentil genotypes studied during 2017–18 and 2018–19. Two varieties Markaz 2009 and Punjab 2009 were used as checks.**
(DOCX)

**S2 Table. Mean±SE for quantitative traits of 5% best performing lentil genotypes during 2017–18.**
(DOCX)

**S3 Table. Mean±SE for quantitative traits of 5% best performing lentil genotypes during 2018–19.**
(DOCX)

**S4 Table. Principal component analysis for quantitative traits of lentils studied during 2017–18.**
(DOCX)

**S5 Table. Principal Component Analysis for quantitative traits studied during 2018–19.**
(DOCX)

## Acknowledgments

We are grateful to Bio-Resource Conservation Institute (BCI), NARC, Islamabad, Pakistan for providing Lentil germplasm and experimental field facilities. The authors highly acknowledge the assistance by Dr. Danish Ibrar, SSO, Oilseeds Research Program and Dr. Shahbaz Khan, SSO, Land Resources Research Institute, NARC, Islamabad for their inputs for improving the manuscript.

## Author Contributions

**Conceptualization:** Muhammad Sajjad Iqbal.

**Data curation:** Syed Atiq Hussain, Muhammad Kashif Ilyas.

**Formal analysis:** Syed Atiq Hussain, Muhammad Akbar.

**Investigation:** Noshia Arshad.

**Methodology:** Noshia Arshad, Saba Munir.

**Project administration:** Muhammad Sajjad Iqbal.

**Resources:** Saba Munir, Tahira Ahmad, Nazra Shaheen, Ayesha Tahir, Muhammad Ahson Khan.

**Software:** Muhammad Azhar Ali.

**Validation:** Hajra Masood, Muhammad Kashif Ilyas.

Writing – **original draft:** Syed Atiq Hussain, Muhammad Sajjad Iqbal.

Writing – **review & editing:** Abdul Ghafoor.

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
