## [Decision Letter · Decision Letter 0]

5 Aug 2021

PONE-D-21-20067

Estimating genetic variability among diverse lentil collections through novel multivariate techniques

PLOS ONE

Dear Dr. Iqbal,

Thank you for submitting your manuscript to PLOS ONE. After careful consideration, we feel that it has merit but does not fully meet PLOS ONE’s publication criteria as it currently stands. Therefore, we invite you to submit a revised version of the manuscript that addresses the points raised during the review process.

We look forward to receiving your revised manuscript.

Kind regards,

Tzen-Yuh Chiang

Academic Editor

PLOS ONE

“There is no financial assistance for publication of this research article”

Reviewers' comments:

Reviewer's Responses to Questions

**Comments to the Author**

1. Is the manuscript technically sound, and do the data support the conclusions?

Reviewer #1: No

Reviewer #2: Yes

2. Has the statistical analysis been performed appropriately and rigorously? 

Reviewer #1: No

Reviewer #2: Yes

3. Have the authors made all data underlying the findings in their manuscript fully available?

Reviewer #1: Yes

Reviewer #2: Yes

4. Is the manuscript presented in an intelligible fashion and written in standard English?

Reviewer #1: No

Reviewer #2: No

5. Review Comments to the Author

Reviewer #1: In this study, the authors evaluated 220 lentil collections for various quantitative and qualitative traits using augmented design in 2018 and 2019. However, sufficient detail of the field layout and statistical analysis of the raw data was not provided. How many blocks were used? Each genotype was grown on a 5m row but it is not indicated if all rows were arranged in a single block or multiple blocks were used? Augmented design is used when the number of entries are large. Entries are unreplicated but checks should be replicated and are used to adjust entry means for soil gradient differences. The authors did not indicate if they used adjusted means. This is critical because some of the quantitative traits are highly influenced by the environment and the observed differences may be due to the location effect.

The background statement states that low yield of lentil in Pakistan is mainly because of adaptation of low yielding local genotypes in different environments due to low genetic variability. However, the best genotypes with the highest seed yield, early maturity, and lowest cooking time are from Pakistan which contradicts the first statement. Overall, I found several grammatical and punctuation errors which made reading the paper difficult. I advise the authors to address the comments below and resubmit it to local/regional journals as it may be more useful to lentil breeders in and around Pakistan.

Lines 31-32: if (Pak) stands for Pakistan then write in full to make it consistent with the one writes on Line 30.

Lines 37-39: In conclusion, 12 genotypes including 5664, 5687,… Markaz 2009, and Punjab 2009 were selected on …..

Line 42: Lens culinaris Medik.

Line 43: It was domesticated

Line 45: throughout the Mediterranean Basin

Line 46: Lentil is the 4th major pulse crop in the world after common bean, pea, and chickpea, and includes seven taxa and four species such as L. culinaris,

Line 50: Lentils are consumed due to their third-highest level

Line 62: Why is the reference in superscript as opposed to within square brackets? Be consistent throughout.

Lines 67-69: The sentence “For combating effects of biotic and abiotic stresses by….cautiously practicable [12]” is not clear what the authors are trying to say. These are practicable. Are the authors referring to Pakistan? Moreover, reference no 12 was cited here but it has nothing to do with the sentence. Please double check if it is the right reference?

Line 79: quantitative and qualitative traits [20].

Line 97: I suggest moving Table 1 into supplementary material.

Line 100: during 2018 and 2019. It is known that these are two consecutive years. Revise similar expressions throughout.

Line 101: Presentation of Figure 1 in material and methods section is unnecessary. Here describe methods only Eg. Monthly temperatures during these years were recorded at the experimental station. Present the figure in the results section.

Line 101: Reference 16 is superscript again???

Line 125: color of testa were observed???

Table 1: Table caption should be placed above the table and figure caption should be below the figure. This should be supplemental file. Do not present tables and figures in Materials and methods section.

Statistical analysis: XLstat 2019 was used to analyze quantitative and qualitative traits data” The method is not detailed enough. What type of analysis performed? The authors used augmented design with two checks interspersed every 20 lines? But in the statistical analysis there is no mention if entry values were adjusted relative to the checks. Proper ANOVA need to be performed for quantitative traits. Are the frequency distribution, Pearson correlation, and PCA performed based on adjusted entry means??? How is Heritability estimated? Is it broad-sense or narrow-sense? Which formula was used to estimating heritability? It appears that the authors divided VG/VP but there is no mention how VG is computed. The trial was conducted for 2 years but no mention of how this computation was performed. These are critical questions that need to be addressed but the methods lack sufficient detail to assess if the right approach was followed.

Results: Presentation of results should focus on key findings. The authors do not need to repeat everything that can be easily seen in tables. Moreover, if proper ANOVA was conducted and there was no significant effect of year then the two-year means can be presented instead of showing each year data separately which made the paper unnecessarily lengthy.

Table 2: What is V? SP is shown in the footnote but not indicated in the table?? What is VC? Do you mean CV? The authors should have taken time to clean up such mistakes before submitting this manuscript.

Table 6 and 7: As these are qualitative traits, they are almost similar between the two years. Why is it necessary to show almost identical data in two tables?

Tables 9 and 10: What are Fs? Each table should be standalone and descriptions of acronyms should be added in footnotes. Same for Table 11.

Table 11: The authors are trying to predict yield using cooking time as independent variable which doesn’t make sense. How is cooking time contributing to yield? Also they presented correlation of agronomic traits with cooking time. Correlation does not imply causation.

Page 24-24: “Values less than 1.00……. individual differences”. This sentence is not accurate. Heritability is a dynamic number and values less than 1 does not necessarily mean genes did not contribute to the phenotypic variance.

Page 26 line 30, Page 27 line 1: different in text citation. Be consistent and refer to the journal citation and reference formats. There are several other similar cases and need revision throughout.

Reviewer #2: The paper by Hussain et al. focus on the important topic of legume germplasms characterization to feed international breeding programs and promote technological advances in plant alternatives to the 5 major crops in which the World currently relies for protein and carbohydrates provision.

The quality of the english writing is very poor, starting from the first sentence of the abstract (not abstracts as in L21) and with repeated writing problems such as misplaced capital letters and words lacking. This requires professional english correction.

The amount of Tables and Figures is too dense, and authors should revise information to be put as supplementary data or presented in alternative ways. This is also true for the paper in general, I believe that authors must revise the content and make the paper more straightforward and lighter, since it is too extensive in its current form.

In general the reference section has older papers and has to be updated.

L62,101; P31 - reference in the wrong format.

L76 - more recent references, such as, DOI: 10.1002/ppp3.10158; https://doi.org/10.3390/foods9040400

L82 - delete 'in the near past'; this is a current topic

L94 - Change Plants to Plant

L103 - these two check varieties are already described for which traits?

L105 - soil characterization is required

From page 8 forward there is no numbering of the lines.

In the results section, grams appear after the numbers both without spacing or between parenthesis or in full word (e.g p11). The correct form is after the number with spacing e.g. 3.2 g. Correct throughout the whole text.

P9 'Lentils characterized during first year showed better performance as compared to subsequent year.' this is discussion, not results description.

P11 - replace & with and

Table 4/5 are not readable and the authors should present this in an alternative, more friendly reader way. In the titles is not best performed, but best performing

P16/18/22/24 - no need to refer the program again, it's in M&M

Table 9 & 10 can be supplementary data and the results description regarding PCA should focus only on the first two components.

P26 - Jaleel et al, 2009 is not in the correct form

P27 - Pushpavalli et al, 2014 also

P27 - please correct the sentence 'Another aspect of lentils is lowest cooking time considered a favorite trait for consumers.in the current studies, lowest cooking time 10 mint' from punctuation to words misspelling

The discussion on seed coat coloring and the genetic background is quite interesting but I feel it lacks integration in the context of the author's study, as they did not look at this dimension.

The definition of PCA is excusable. This is for academic partners who know this method.

P29 - And not only in Pakistan or Turkey, for example: https://doi.org/10.1071/CP18205

P30 - Once again, punctuation errors should be covered throughout the entire document: 'Moreover, other traits like number of pods. Lower pod height and plant height were also placed in F3, F4, and F5, respectively.'

In the conclusion section, authors shouldn't just summarize the main findings again, that is for the Abstract section; here, the main findings should be commented on further, with the industry and farmers sector perspective, showing the importance of the study for them. Would the authors advise farmers to just use the best performing genotype and forget the rest? Are these 12 genotypes well adapted to different edaphoclimatic regions, or just in the research center site?

6. PLOS authors have the option to publish the peer review history of their article (what does this mean?). If published, this will include your full peer review and any attached files.

Reviewer #1: **Yes: **Teketel Haile

Reviewer #2: No

---

## [Author Response · Author response to Decision Letter 0]

2 Oct 2021

Reviewer #1

1; In this study, the authors evaluated 220 lentil collections for various quantitative and qualitative traits using augmented design in 2018 and 2019. However, sufficient detail of the field layout and statistical analysis of the raw data was not provided. How many blocks were used?

The detail of field layout and statistical analysis is added as mentioned below; 

Eleven blocks were used in this study. Each block consisted of 20 genotypes and two check varieties, viz., Markaz 2009 and Punjab 2009 were planted after every twenty lines. For statistical analysis ten plants were sampled radomly from each line/genotypes. Morphological and agronomical traits were recorded on indvidual plant basis during 2018 and 2019 [16]. Recorded data of agronomical traits were averaged and mean data were subjected to analyse for descriptive statistics, frequncy distribution, Pearson correlation, PCA, cluster pattern analysis, structure equational modling. While broad sense heritablity was estimated by considereing 2018 as replication 1 and 2019 as replication 2. The descriptive statistics including minimum values, maximum values, range, mean, variances, standered deviation,, cofficient of variances and standered error of mean were estimated through statistical softwere (XLStat 2019). 

As the qualitative traits did not changed from both years so combined frequency distribution were estimated [16]. To find the relationship among agronomical traits Pearson correlation was computed with XLStat 2019, as well for PCA. Dendrograms were constructed based on Eucladian distance through RStudio. The structural equation modling was drawn through LISREL (LISREL Student Version 9.2https://lisrel-for-windows-student.software.informer.com/9.2/) to estimate multiple regression that was indicating contribution of quantitative traits toward yield. The genetic and phenotypic variances and broad sense heritabity were estimated by formmula suggested by Singh and Chaudhary [51] through online source PBSTAT (http://pbstat.com/ppb/). 

2; Each genotype was grown on a 5m row but it is not indicated if all rows were arranged in a single block or multiple blocks were used? 

Eleven blocks were used in this study. Each block consisted of 20 genotypes and two known cultivars, viz., Markaz 2009 and Punjab 2009 were planted as check (control) after every twenty lines.

3;Augmented design is used when the number of entries are large. Entries are unreplicated but checks should be replicated and are used to adjust entry means for soil gradient differences. The authors did not indicate if they used adjusted means. This is critical because some of the quantitative traits are highly influenced by the environment and the observed differences may be due to the location effect.

The adjusted mean values were used. Sentence is added in statistical analysis for clarification about adjusted mean and soil was homogenously distributed as we conducted and provided soil analysis (Line 113). 

4;The background statement states that low yield of lentil in Pakistan is mainly because of adaptation of low yielding local genotypes in different environments due to low genetic variability. However, the best genotypes with the highest seed yield, early maturity, and lowest cooking time are from Pakistan which contradicts the first statement.

Before collection, evaluation and characterization there were no prior information about yield potential. After conducting subject mentioned analysis we come to know that current germplasm representing best performance is mainly based on collections belonged to Pakistan. We assumed outcome of this trial will lead to explore them under different environments for future varietal development. 

5; Overall, I found several grammatical and punctuation errors which made reading the paper difficult. I advise the authors to address the comments below and resubmit it to local/regional journals as it may be more useful to lentil breeders in and around Pakistan.

We rectified the points raised, our study is of international significance because of germplasm collection representing USA, Syria, Pakistan, and few are of unknown origin. Therefore it has scope of international use and wide readability.

6;Lines 31-32: if (Pak) stands for Pakistan then write in full to make it consistent with the one writes on Line 30.

We replaced the word Pak with Pakistan throughout text 

7;Lines 37-39: In conclusion, 12 genotypes including 5664, 5687,… Markaz 2009, and Punjab 2009 were selected on …..

In conclusion, 12 genotypes including 5687, 6084, 6062, 6122, 6058, 6087, 5689, 6042, 6074, Markaz 2009 & Punjab 2009 were selected on the basis of best performance especially seed yield to other traits viz., biological yield, plant height, pod size, cooking time and maturity time etc. 

8;Line 42: Lens culinaris Medik. 

Full stop is added at the end of Medik.

9;Line 43: It was domesticated

Changed as suggested 

10;Line 45: throughout the Mediterranean Basin

‘The’ is included before Mediterranean basin 

11;Line 46: Lentil is the 4th major pulse crop in the world after common bean, pea, and chickpea, and includes seven taxa and four species such as L. culinaris,

Sentence revised as desired

12;Line 50: Lentils are consumed due to their third-highest level

Lentil is consumed 

13;Line 62: Why is the reference in superscript as opposed to within square brackets? Be consistent throughout.

Corrected according to format of Plos One references 

14;Lines 67-69: The sentence “For combating effects of biotic and abiotic stresses by…. cautiously practicable [12]” is not clear what the authors are trying to say. These are practicable. Are the authors referring to Pakistan? 

Corrected as;

It is not economically possible to reduce biotic stressors by utilizing pesticides and herbicides, and to reduce abiotic stresses by supplementing soils to meet plant demands through reclamations and enhancing fertiliser, as well as by improving irrigation systems [13].

15;Moreover, reference no 12 was cited here but it has nothing to do with the sentence. Please double check if it is the right reference?

Amended, the correct reference is [13]

16;Line 79: quantitative and qualitative traits [20].

Corrected 

17;Line 97: I suggest moving Table 1 into supplementary material.

Moved as supplementary material 

18;Line 100: during 2018 and 2019. It is known that these are two consecutive years. Revise similar expressions throughout.

Revised: during 2018 and 2019

19;Line 101: Presentation of Figure 1 in material and methods section is unnecessary. Here describe methods only Eg. Monthly temperatures during these years were recorded at the experimental station. Present the figure in the results section.

Figure now moved to results section (P17).

20;Line 101: Reference 16 is superscript again???

Corrected: [16]

21;Line 125: color of testa was observed???

Corrected 

22;Table 1: Table caption should be placed above the table and figure caption should be below the figure. This should be supplemental file. Do not present tables and figures in Materials and methods section.

Amended as desired. 

23;Statistical analysis: XLstat 2019 was used to analyze quantitative and qualitative traits data” The method is not detailed enough.

Detail is provided now

25; What type of analysis performed? 

Recorded data of agronomical traits were averaged and mean data were subjected to analyse for identification of best genotypes, descriptive statistics, frequency distribution, pearson correlation, PCA, hiererchical clster analysis, structure equational modling and broad sense heritablity.

26;The authors used augmented design with two checks interspersed every 20 lines? But in the statistical analysis there is no mention if entry values were adjusted relative to the checks.

These values are adjusted values as detail is given at point 1.

27; Proper ANOVA need to be performed for quantitative traits. Are the frequency distribution, Pearson correlation, and PCA performed based on adjusted entry means??? 

ANOVA performed and values of significant traits added in Table 2. Yes, the frequency distribution, Pearson correlation, and PCA performed based on adjusted entry means for each year. 

28; How is Heritability estimated? 

It is estimated from mean values of traits for each genotype during both years.

29; Is it broad-sense or narrow-sense? 

It is broad sense heritability

 30; Which formula was used to estimating heritability? It appears that the authors divided VG/VP but there is no mention how VG is computed.

Singh and Chaudhary [28] method was followed as provided below 

Where GV and PV is the genetic variances and phenotypic variances respectively gms= genetic mean square

ems= error mean square

r= number of replications

EV= envirenmental variances 

h2= broad sens heritability 

References used for heritability [29] 

Popat R, Patel R, Parmer D. Variability: Genetic variability analysis for plant breeding research. R package version. http://cran.rproject.org/web/packages/variability/index.html. 2020. Accessed 07 September 2021.

We revised whole table by performing analysis again and showing more parameters to increase understanding of the data.

31; The trial was conducted for 2 years but no mention of how this computation was performed. These are critical questions that need to be addressed but the methods lack sufficient detail to assess if the right approach was followed.

Included following information in the main text as already mentioned under point 1; 

Ten plants were labeled as sample from each genotypes and agronomical traits were recorded on indvidual plant basis during 2018 and 2019 [16] Recorded data of agronomical traits were averaged and mean data were entered into excel 2016 separatly for 2018 and 2019. Both years data was separatly subjected to analyse for identification of best genotypes, descriptive statistics, frequncy distribution person correlation PCA, dendrogram, structure equational modling while broad sens heritablity was estimated by considereing 2018 as replication 1 and 2019 as replication 2. The descriptive statistics including minimum values, maximum values, range , mean, variances, standered deviation,, cofficient of variancecs and standered error of mean were estimated through statistical softwere (XLstat 2019). As the qualitative traits did not changed from both years so combined frequency distribution for both years were estimated [16]. To find the relation ship among agronomical traits person correlation were computed with XLstat 2019, same staitiscal tool was used for PCA. Dendrogram was constracted based on eucladian distance through RStodio. The structural equation modle was constrcuted through LISREL (LISREL Student Version 9.2 https://lisrel-for-windows-student.software.informer.com/9.2/) to estimate multiple regression that was indicating contribution of quantitative traits toward yield. 

32; Results: Presentation of results should focus on key findings. The authors do not need to repeat everything that can be easily seen in tables. 

Done according to suggestion

33; Moreover, if proper ANOVA was conducted and there was no significant effect of year then the two-year means can be presented instead of showing each year data separately which made the paper unnecessarily lengthy.

ANOVA performed and found significant effects which are now included in Table 2. 

34; Table 2: What is V? SP is shown in the footnote but not indicated in the table??

Corrected: V is variance. 

SP is present in row number 7

 35; What is VC? Do you mean CV? 

Corrected, yes, it is CV 

The authors should have taken time to clean up such mistakes before submitting this manuscript.

Rectified mistakes as advised throughout the text.

36; Table 6 and 7: As these are qualitative traits, they are almost similar between the two years. Why is it necessary to show almost identical data in two tables?

Table 6 removed 

37; Tables 9 and 10: What are Fs? Each table should be standalone and descriptions of acronyms should be added in footnotes. Same for Table 11.

F represents factors 

Description of acronyms added in footnotes for table 9 and 10 

38; Table 11: The authors are trying to predict yield using cooking time as independent variable which doesn’t make sense. How is cooking time contributing to yield? they presented correlation of agronomic traits with cooking time. Correlation does not imply causation.

Agreed with respectable Reviewer and we removed cooking time 

39; Page 24-24: “Values less than 1.00……. individual differences”. This sentence is not accurate. Heritability is a dynamic number and values less than 1 does not necessarily mean genes did not contribute to the phenotypic variance.

Agreed, heritability analysis is revised after excluding qualitative traits. Few more statistical parameters like genetic advance and genetic gain are included.

40; Page 26 line 30, Page 27 line 1: different in text citation. Be consistent and refer to the journal citation and reference formats. There are several other similar cases and need revision throughout.

Corrected page 26 line 30 and page 27 line 1. Revised throughout the text

 

Reviewer #2: 

The paper by Hussain et al. focus on the important topic of legume germplasms characterization to feed international breeding programs and promote technological advances in plant alternatives to the 5 major crops in which the World currently relies for protein and carbohydrates provision.

The quality of the English writing is very poor, starting from the first sentence of the abstract (not abstracts as in L21) and with repeated writing problems such as misplaced capital letters and words lacking. This requires professional english correction.

English language corrections and editing is performed by local English expert, Dr. Behzad Anwar, Department of English, UOG 

The amount of Tables and Figures is too dense, and authors should revise information to be put as supplementary data or presented in alternative ways. This is also true for the paper in general, I believe that authors must revise the content and make the paper more straightforward and lighter, since it is too extensive in its current form.

Authors consider suggestions, removed extra text and moved the tables as supplementary material 

In general the reference section has older papers and has to be updated.

New and updated references are included now 

L62,101; P31 – reference in the wrong format.

Corrected according to format 

L76 - more recent references, such as, DOI: 10.1002/ppp3.10158; https://doi.org/10.3390/foods9040400

Suggested reference added 

L82 - delete 'in the near past'; this is a current topic

Deleted 

L94 - Change Plants to Plant

 Changed 

L103 - these two check varieties are already described for which traits?

Markaz 2009; suitable for barani and irrigated areas of the plains and Pothwar region. Fimilar as high yielding, drought tolerant and lodging resistant.

Punjab 2009; descried for high yield, moderately resistant to rust and stem rot, tolerant to lodging

L105 - soil characterization is required

Soil type used was analysed by Soil Science Laboratory, National Agricultural Research Centre (NARC), Islamabad and was Nabipur series, coarse loamy, mixed, hyperthemic, and udic ustochrepts. Added at Line 113.

From page 8 forward there is no numbering of the lines.

We make text continue with line numbering 

In the results section, grams appear after the numbers both without spacing or between parenthesis or in full word (e.g p11). The correct form is after the number with spacing e.g. 3.2 g. Correct throughout the whole text.

Corrected throughout text.

P9 'Lentils characterized during first year showed better performance as compared to subsequent year.' this is discussion, not results description.

P11 - replace & with and

Moved to discussion section; Lentils characterized during first year showed better performance as compared to the subsequent year.

Table 4/5 are not readable and the authors should present this in an alternative, more friendly reader way. In the titles is not best performed, but best performing

Revised as suggested. 

P16/18/22/24 - no need to refer the program again, it's in M&M

Revised 

Table 9 & 10 can be supplementary data and the results description regarding PCA should focus only on the first two components.

Table 9 and 10 moved to supplementary data. The result description of PCA revised according to suggestion.

P26 - Jaleel et al, 2009 is not in the correct form

Corrected 

P27 - Pushpavalli et al, 2014 also

Corrected 

P27 - please correct the sentence 'Another aspect of lentils is lowest cooking time considered a favourite trait for consumers.in the current studies, lowest cooking time 10 mint' from punctuation to words misspelling

Corrected 

The discussion on seed coat coloring and the genetic background is quite interesting but I feel it lacks integration in the context of the author's study, as they did not look at this dimension.

Appropriate text has been included 

The definition of PCA is excusable. This is for academic partners who know this method.

Unnecessary text has been removed 

P29 - And not only in Pakistan or Turkey, for example: https://doi.org/10.1071/CP18205

Included 

P30 - Once again, punctuation errors should be covered throughout the entire document: 'Moreover, other traits like number of pods. Lower pod height and plant height were also placed in F3, F4, and F5, respectively.'

Rectified 

In the conclusion section, authors shouldn't just summarize the main findings again, that is for the Abstract section; here, the main findings should be commented on further, with the industry and farmers sector perspective, showing the importance of the study for them. Would the authors advise farmers to just use the best performing genotype and forget the rest? 

Conclusion revised as suggested. 

Are these 12 genotypes well adapted to different edaphoclimatic regions, or just in the research center site?

These 12 genotypes were selected on best performance out of 220 after conducting two years of evaluation and characterization under same edaphoclimatic conditions therefore recommended for the area having similar features like Photohar, arid and semi-arid regions. Moreover, these genotypes have potential to become stable varieties after trials.

---

## [Decision Letter · Decision Letter 1]

8 Nov 2021

PONE-D-21-20067R1Estimating genetic variability among diverse lentil collections through novel multivariate techniquesPLOS ONE

Dear Dr. Iqbal,

Thank you for submitting your manuscript to PLOS ONE. After careful consideration, we feel that it has merit but does not fully meet PLOS ONE’s publication criteria as it currently stands. Therefore, we invite you to submit a revised version of the manuscript that addresses the points raised during the review process.

We look forward to receiving your revised manuscript.

Kind regards,

Tzen-Yuh Chiang

Academic Editor

PLOS ONE

Reviewers' comments:

Reviewer's Responses to Questions

**Comments to the Author**

1. If the authors have adequately addressed your comments raised in a previous round of review and you feel that this manuscript is now acceptable for publication, you may indicate that here to bypass the “Comments to the Author” section, enter your conflict of interest statement in the “Confidential to Editor” section, and submit your "Accept" recommendation.

Reviewer #1: All comments have been addressed

Reviewer #2: (No Response)

2. Is the manuscript technically sound, and do the data support the conclusions?

Reviewer #1: Yes

Reviewer #2: Yes

3. Has the statistical analysis been performed appropriately and rigorously? 

Reviewer #1: Yes

Reviewer #2: Yes

4. Have the authors made all data underlying the findings in their manuscript fully available?

Reviewer #1: Yes

Reviewer #2: Yes

5. Is the manuscript presented in an intelligible fashion and written in standard English?

Reviewer #1: No

Reviewer #2: No

6. Review Comments to the Author

Reviewer #1: The revised manuscript has been greatly improved but there are still some grammatical and punctuation issues. Please address my comments attached.

Reviewer #2: The authors have addressed some of the comments, however, there are still inconsistencies and issues to be resolved.

Regarding the English, although the authors have improved and revised the manuscript, there are still several problems which make the text not correct. Most importantly, the Abstract must be completely revised as it is not written in correct english.

Furthermore, the discussion section is too dense. It must be more focused and concise.

In the previous revision, I had asked for the authors to re-write the Conclusion section, being more critical about the importance of the study instead of just summarizing the results. Some text was changed, but it still continues very focused on the summary of results.

Please note that numbers below 10 must be written in full. Correct throughout the entire manuscript.

L70: the lack of adaptation? the difficulty to adapt?

L139/L146: please correct Hierarchical Cluster Analysis

L147: correct drwan to drawn

L171: correct the number of the means in parenthesis

L180: missing one dot at the end of the sentence

L181: why is the unit (g)? Again there are still inconsistencies in the unit presentation; in P11 grams appear in full; with minutes is the same; sometimes appear in full, others as min. or min - please homogenize through the entire document.

L181-185: please revise the punctuation of this sentence, as it is now it is not understandable

L187: significantly contributes?

No line numbering after Page 9.

P11: remained insignificant does not mean anything - please correct;

P12: The first sentence of is not comprehensible. Re-write.

P13: The parameters analyzed sometimes appear as abbreviation and other times in full, e.g. biological yield / BY

P15: There is some repeated text, for example, "The factor 1 component contributed 19.05% total variation and factor 2 as 12.51%." appears twice

P24: Again, this sentence is irrelevant, please delete. "PCA is an approach used statistically to distribute number of variables in data by extracting important one from a wide number of genotypes."

P24: Discussion on PCA factors besides 1 and 2 is irrelevant and should be deleted.

7. PLOS authors have the option to publish the peer review history of their article (what does this mean?). If published, this will include your full peer review and any attached files.

Reviewer #1: No

Reviewer #2: No

---

## [Author Response · Author response to Decision Letter 1]

18 Jan 2022

Response to Reviewer’s comments

Abstract 

L25 The numbers don’t add up. It is 224 instead of 220??

Current collection is of 220 genotypes representing Pakistan (178), Syria (14), USA (22) and 6 are of unknown origin. Exact number included throughout the text. 

L36: Define the acronym NP at first mention. Acronyms should be defined at first mention and used consistently. Example: CT was defined on L29 but “cooking time” used on L33 and “CT” on L34. Be consistent!

Corrected as advised.

L37-38: Revise “The PCA revealed 11 components, 38 make convenience selection for individual components for future hybridization” 

We revised the text as suggested and excluded that deem unfit.

The PCA revealed a considerable reduction in components for selection of suitable genotypes with desired traits that be utilized for future lentil breeding.

L40: …showed a perfect relationship OR ..showed that the relationship is perfect.

Revised as commented. 

Structural Equational Model (SEM) for SY based on covariance studies indicated the perfect relationship among variables.

Introduction

L50-53: “Lentil is the 4th major pulse crop in the world after common bean, pea, and

chickpea, and includes seven taxa with four species such as L. culinaris, L. nigricans, L.

ervoides, and L. lamottei, having two varietal types as macro-sperma (large seeded) and micro-sperma (small seeded) [4,5].” Revise this sentence which states that all four species have two varietal types but only the cultivated culinaries has two types. The others are wild not cultivated. 

Sentence revised as like following

Lentil is the 4th major pulse crop in the world after common bean, pea, and chickpea, and included seven taxa with four species such as L. culinaris, L. nigricans, L. ervoides, and L. lamottei, the only cultivated species (L. culinaris) have two varietal types as macro-sperma (large seeded) and micro-sperma (small seeded) [4,5].

L62: developing countries

Corrected 

L71: Economically feasible is better expression

Replaced.

It is not economically feasible to reduce biotic stressors by utilizing pesticides, herbicides and to reduce abiotic stresses by supplementing soil to meet plant demand through reclamation and enhancing fertilizer, as well as by improving irrigation systems [13].

L86: Therefore, the objective of this study was to assess…

Text revised as stated as;

The current studies used several statistical techniques to unwind the hidden potential of genetic diversity in the available lentil genetic resources and to assess variability in agro-morphological traits for consecutively two years. Identified novel genotypes would be the base material for future lentil breeding program.

Materials and Methods

L98: Two hundred and twenty-four lentil

Corrected as the number is of 220 and replaced throughout the text where applicable. Ambiguity of 224 is removed.

L106: Experiments were laid out in an augmented design. Not lentil seeds!!

Corrected.

Experiments were laid out in an augmented design within 11 blocks and each block consisted of 20 genotypes along with two check varieties, viz., Markaz 2009 and Punjab 2009 after every twenty lines [16].

L111: Find a better word than “unwinding”. May be studied 

Revised as; actual genetic diversity may be studied.

The trial was conducted under usual agricultural practices without applying any herbicide, fungicide, pesticide, or fertilizer so that the actual genetic diversity may be studied.

L122: and hard seeds (HS) were counted after boiling 100 seeds.

Correction made as per suggestion. 

After harvesting, biological yield (BY), seed yield (SY), 100-seed weight (100-HSW), cooking time (CT), and hard seeds (HS) were counted after boiling 100 seeds.

L133: define 0 to 4 and 1 to 3 scales similar to flower ground color above. 

Well defined all of the traits accordingly. 

At late vegetative stage, SSP (rated as 0; absent and 1; present), LS (3; small, 5; medium, and 7; large) and LP (0; absent, 3; slight, and 7; dense) were observed. At early reproductive stage, GCF (1; white color, 2; white with blue lines, 3; blue, 4; violet, 5; pink, and 6; other) was observed. At late reproductive stage, PP (0; absent and 1; present) was observed. At maturity stage, PD (0; none, 3; low, 5, medium, and 7; high) and PSh (0; none, 3; low, 5, medium, and 7; high) were surveyed. Tendril length (1; rudimentary tendril and 2; prominent tendril) was also observed during late reproductive stage. After harvesting, CPT (0; absent, 1; olive, 2; grey, 3; brown, 4; black.), CC (1; yellow color, 2; orange-red, and 3; olive green) and GCT (1; green, 2; grey, 3; brown, 4; black, and 5; pink) was observed. 

L151 change Via through R Studio to using R Studio: Via means through.

Changed and the whole paragraph is revised in view of the suggestion. 

using R Studio

Results

L180: full stop after 2.71 g.

Sentence paragraphed revised. 

SY in 2018-19 as compared to SY of 20117-18. Seed yield ranged from 5 to 85 g in 2017-18 and 3 to 76 g during 2018-19, respectively. Likewise, 6084 belonged to Narowal, Pakistan produced the highest SY in both years. The lowest SY was recorded in genotype 5475 originated from Gujranwala, Pakistan during 2017-18, whereas the genotype 5494 (Okara, Pakistan) and 5475 (Gujranwala, Pakistan) were the lowest in SY during 2018-19. Genotypes 6052 (Syria), 5583 (Muzaffargarh, Pakistan), and 5556 (Hyderabad, Pakistan) produced better SY during both years. Overall performance of genotypes was better during the first year as compared to the subsequent year that could be attributed due to better environmental conditions during second year, however, genotypic trend remained similar during both years indicating evidence of heritable heritability among genotypes.

L182: change the expression “was ranged” to “ranged” throughout the text

Changed throughout the text as suggested. 

L205: change the expression “both the years” to “both years” throughout the text.

Revised according to suggestion. 

L215: Where is Faisalabad? In Pakistan?

Yes, Faisalabad is in Pakistan and Pakistan included in the text along the cities representing as origin. 

Use of grams and g throughout the paper be consistent!

g used throughout the text. 

Page 12 L1: which can be used to describe lentil on…

Revised accordingly 

Page 13: Like, hundred seed weight exhibited… Similarly, hundred seed weight exhibited

Sentence revised accordingly 

The 100-SW was positively correlated with BY, while BY was positively correlated with PH.

Page16: The paragraph below Fig. 3 caption needs revision for grammatical correctness. 

Example: scattered plots were (…..)

 indicating a way to screen genotypes 

 therefore, increase in height would result in higher number of seeds per pods 

 that can help in selection for genotypes

The whole paragraph revised accordingly 

Genotypes (5506, 5729, 5510, 5672, 5636, 5581, 5491, 5483) at the upper right corner were grouped on the basis of 100-SW, BY and SY indicating productive genotypes while in lower right corner genotypes (5472, 5512, 5479, 5486, 5650, 5600, 5531) reflected variation due to PH, NP, and HS indicating close association of PH with NP and HS. Therefore, increase in height would result in higher number of seeds per pods that can help in selection for genotypes. Similarly, at the upper left side of the scattered plot, genotypes (6080, 6043, 5653, 6043, 6002, 6060, and 5677) showed varying responses to CT, SP, and PS that can help in selection for genotypes with desirable CT contributed by NSP and PS which affects possible CT. While on the other hand, in lower left-side genotypes 23787, 24787, 23779, 5742, and 5712 were placed with the highest variability in DM and LPH (Fig. 4). It can infer that DM has a significant impact on LPH.

Page 17: The authors keep talking about 220 lines but there are 184 Pakistan +14 Syria + 22 USA +04 Unknown with a total of 224 lines. Where did the 4 lines go? It’s really surprising the authors don’t know the exact number of lines in their 2 year study!

The exact number of collection is 220 as stated in point 1. Misperception was due to typographic error. We apologize for misunderstanding. 

What does C2bii mean on page 17?

C2bii represented the sub-cluster under cluster C2. The rephrased the sentences to avoid complication of the figure. 

Cluster C1 consisted of 37 genotypes while cluster C2 consisted of 115 genotypes. Cluster C3 consisted of 12 genotypes whereas clusters C4 and C5 consisted of only 1 (24783) and 4 genotypes (6084, 6058, 6062, and 6122) respectively. Cluster C6 consisted of 49 genotypes and cluster C7 consisted of 2 genotypes. Best performed genotypes were considered in C3 during 2017-18 and C5 during 2018-19 due to their performance as SY, BY, PS, NSP, CT, and HS. Cluster C2 belongs to 100-SW >1.19 producing genotypes during 2017-18 while those genotypes which produced 100-SW >1.19 during 2018-19 were grouped into cluster C3. Cluster C2 also consisted of those genotypes which had PH >55.1 and DM >170 in both years. Cluster C1 consisted of genotypes having SY > 40 or near to 40g during both years. 

Page17: Cluster C7 consisted of only 2 genotypes (5664 Layyah and 6074 Bahawalnagar) which belong to Pakistan. Best performing genotypes were considered in C4 during 2018 and C5 during 2019 due to their performance. ). 

Sentence revised accordingly 

Best performed genotypes were considered in C3 during 2017-18 and C5 during 2018-19 due to their performance as SY, BY, PS, NSP, CT, and HS. 

Page 18: Structural Equational Model (SEM): I still don’t get it how cooking time and hard seed count after cooking influence yield? The statistical software are garbage in garbage out. The authors should be cautious in making sense of the data. This doesn’t make sense!

SEM analysis revised to make uniformity of the data while cooking time removed as suggested.

Table 5. Structural equations of model representing regression for agronomic traits studied during 2017-18 and 2018-19. 

SY 100SW BY PH LPH NSP PS DM NP

Z-values 2.12 0.27 0.02 0.43 3.69 5.43 -0.04 2.21

P-values 0.10 0.09 0.00 0.01 0.81 0.0 0.0 0.04

SY = 2.16*100-SW + 0.27*BY + 0.025*PH + 0.0.43*LPH + 1.826*NSP + 6.34*PS - 0.039*DM + 2.298*NP 

SY, seed yield; BY, biological yield; PH, plant height; LPH, lower pod height; PS, pod size; DM, days to maturity; NP, number of pods; NSP, number of seeds per pod. 

Figure 8. Structural equational model for seed yield (SY)

Page 19: with high genetic advance as percentage of mean 79.71 for 2018. What does this mean?

Following description has been included 

It indicates predominance of additive gene action for these traits under prevailing conditions. Further, high genetic advance along high heritability estimates provides ample opportunity for most suitable selection of traits for future application. 

Page 19: Heritability analysis: “The high heritability was observed for BY (0.99) along with high genetic advance as percentage of mean 79.71 for 2018 while for the subsequent year lower pod height 27.94. High genetic gain 2065.13 for the year 2018 was recorded for SY while low genetic gain 0.07 for LPH. Similarly, BY showed high genetic gain 1638.62 for second year and low 0.07 for LPH.” The authors are talking about genetic gain and genetic advance here and in Table 6 but no mention of how they are calculated in the materials and methods section. Genetic gain is response to selection over time. I don’t think it is the right term here. Please revise!

Agreed with the honorable reviewer’s observation and we have included genetic advance calculation procedure/formula under material & methods section while excluded the genetic gain. 

Discussion

Page 21 L9: Put comma following in our study, throughout

Comma added. 

Page 22: Although water scarcity affects productivity of legumes at any growth stage, drought during reproductive and grain filling stages (terminal drought) are more critical and usually result in a significant loss of grain yield [39]

Sentence revised as suggested above. 

Variation in native, as well as exotic collections, was observed for both years. Although water scarcity affects productivity of legumes at any growth stage, drought during reproductive and grain filling stages (terminal drought) are more critical and usually result in a significant loss of SY.

Page 22 L10-14: In our study, 5698 (Sialkot, Pakistan) and 6015 (USA), were considered as early maturing genotypes with 161-172 days in 2018 while 24783 (unknown) and 5561 (Sanghar, Pakistan) were found early maturing 161-172 days during 2019 which is more appealing as compared to previous reports of 168-190 days (cite the reference of the previous report here).

Overall sentence has been revised and desired reference has been included. 

(16)

Page 22 2nd paragraph: It is evident that seed yield is directly proportional to number of seeds per pod, higher number of seed per pod will result in higher seed yield.

Sentence has been revised accordingly 

It is evident that SY is directly proportional to NSP, and in current studies, NP ranged from 2-3 in 2017-18 and 1-3 in 2018-19. 

Page 22: lowest cooking time 10 mint should be minutes (or m).

Amended. 

Page 23: Grey ground color of testa and orange-red cotyledon color with grey color of pattern on testa was observed in most lines. 

Sentence revised accordingly 

Grey ground color of testa and orange-red cotyledon color with grey color of pattern on testa was observed in majority. Various markets demand specific seed coat colors and/or patterns. We attempted to determine the percentage of color of testa, cotyledon color and color of pattern.

Page 24: use of & instead of and in the text (need to be discussion)

Authors preferred to use and instead of & throughout the text 

Page 25: Revise “It is also infer that presence of genotypes…” not grammatically correct. It can be inferred that…

Sentence has been revised (need to be check it grammatically)

It can be inferred that presence of genotypes having different origins in the same cluster might be close association as common ancestors but on contrary presenting in different clusters also showed variation in genetic material.

Page 26: Most of the agro-morphological traits had fairly higher heritability [18]

Sentence has been revised accordingly 

Most of the agro-morphological traits had fairly higher heritability [18].

---

## [Decision Letter · Decision Letter 2]

7 Feb 2022

PONE-D-21-20067R2Estimating genetic variability among diverse lentil collections through novel multivariate techniquesPLOS ONE

Dear Dr. Iqbal,

Thank you for submitting your manuscript to PLOS ONE. After careful consideration, we feel that it has merit but does not fully meet PLOS ONE’s publication criteria as it currently stands. Therefore, we invite you to submit a revised version of the manuscript that addresses the points raised during the review process.

We look forward to receiving your revised manuscript.

Kind regards,

Tzen-Yuh Chiang

Academic Editor

PLOS ONE

Journal Requirements:

Reviewers' comments:

Reviewer's Responses to Questions

**Comments to the Author**

1. If the authors have adequately addressed your comments raised in a previous round of review and you feel that this manuscript is now acceptable for publication, you may indicate that here to bypass the “Comments to the Author” section, enter your conflict of interest statement in the “Confidential to Editor” section, and submit your "Accept" recommendation.

Reviewer #1: All comments have been addressed

Reviewer #2: All comments have been addressed

2. Is the manuscript technically sound, and do the data support the conclusions?

Reviewer #1: Yes

Reviewer #2: (No Response)

3. Has the statistical analysis been performed appropriately and rigorously? 

Reviewer #1: Yes

Reviewer #2: (No Response)

4. Have the authors made all data underlying the findings in their manuscript fully available?

Reviewer #1: Yes

Reviewer #2: (No Response)

5. Is the manuscript presented in an intelligible fashion and written in standard English?

Reviewer #1: No

Reviewer #2: (No Response)

6. Review Comments to the Author

Reviewer #1: This manuscript has improved a lot from the previous version. The authors managed to address all my previous comments. However, there are still some grammatical and punctuation issues that need to be addressed before proceeding to publication. Please find below some of my comments.

Line 141: remove double comma and adjust spacing.

P11, L 8-9: What do you mean by there in “Short duration genotypes are suggested as early maturing genotypes hence required further trials to see there early maturity cultivar status.” Delete the entire sentence.

P12, L5-7: Remove “Traits which possess low variability must be required to address through new collection by broadening existing lentil genetic resources through local as well as exotic sources.” There are sentences at the end of each paragraph which are not making any sense. In the result section only point out to important findings. Discussion of results should be in the discussion section.

P13, L4-7: Under Pearson correlation matrix: authors indicated that HSW exhibited positive correlation with PS and BY. PH and PS also showed positive correlation with NSP

and DM but Table 4 only shows correlation of SY with other traits in both years. Either you have to show all pairwise correlation in table 4 or don’t write about results that are not shown in the table.

P 21, L 16: Revise the sentence as “In this study, HSW varied between the years with 3.65 g in 2017-18 and 3.28 g in 2018-19”

P21, L 22: Cooking time not only saves energy and time but also it develops ….

P 23, L5: replace & with and

P 23, L 9: Revise as “Similar results were observed”

Conclusion: tons/hectare versus t/h be consistent throughout

Reviewer #2: (No Response)

7. PLOS authors have the option to publish the peer review history of their article (what does this mean?). If published, this will include your full peer review and any attached files.

Reviewer #1: No

Reviewer #2: No

---

## [Author Response · Author response to Decision Letter 2]

28 Feb 2022

Response to Reviewers comments

Authors pay thanks to the honorable Reviewers to improve this article a standard MS for wide readership. Further, MS has again reviewed thoroughly and edited sufficiently to make intangible and according to Standard English.

Below is the response to the Reviewers comments please;

Line 141: remove double comma and adjust spacing.

Double commas havs been removed and spacing adjusted accordingly, now at line 145 of the manuscript 

P11, L 8-9: What do you mean by there in “Short duration genotypes are suggested as early maturing genotypes hence required further trials to see there early maturity cultivar status.” Delete the entire sentence.

Deleted at line 197 of the final manuscript without track changes

P12, L5-7: Remove “Traits which possess low variability must be required to address through new collection by broadening existing lentil genetic resources through local as well as exotic sources.” There are sentences at the end of each paragraph which are not making any sense. In the result section only point out to important findings. Discussion of results should be in the discussion section.

Authors have removed all of the unnecessary paragraphs as advised by the respectable reviewer, above mentioned lines have been removed as at line 220 of the manuscript

P13, L4-7: Under Pearson correlation matrix: authors indicated that HSW exhibited positive correlation with PS and BY. PH and PS also showed positive correlation with NSP

and DM but Table 4 only shows correlation of SY with other traits in both years. Either you have to show all pairwise correlation in table 4 or don’t write about results that are not shown in the table.

Authors have removed the suggested characters and restricted to the traits as recommended. 

P 21, L 16: Revise the sentence as “In this study, HSW varied between the years with 3.65 g in 2017-18 and 3.28 g in 2018-19”

Revised accordingly at line 386

P21, L 22: Cooking time not only saves energy and time but also it develops ….

Amended as suggested

P 23, L5: replace & with and

Replaced accordingly at line 420 and also in other paragraphs

P 23, L 9: Revise as “Similar results were observed”

Revised accordingly as suggested in the line 424 

Conclusion: tons/hectare versus t/h be consistent throughout

Tons/hectare added as suggested in the line 477.

---

## [Decision Letter · Decision Letter 3]

17 Mar 2022

PONE-D-21-20067R3Estimating genetic variability among diverse lentil collections through novel multivariate techniquesPLOS ONE

Dear Dr. Iqbal,

Thank you for submitting your manuscript to PLOS ONE. After careful consideration, we feel that it has merit but does not fully meet PLOS ONE’s publication criteria as it currently stands. Therefore, we invite you to submit a revised version of the manuscript that addresses the points raised during the review process.

We look forward to receiving your revised manuscript.

Kind regards,

Tzen-Yuh Chiang

Academic Editor

PLOS ONE

Journal Requirements:

Reviewers' comments:

Reviewer's Responses to Questions

**Comments to the Author**

1. If the authors have adequately addressed your comments raised in a previous round of review and you feel that this manuscript is now acceptable for publication, you may indicate that here to bypass the “Comments to the Author” section, enter your conflict of interest statement in the “Confidential to Editor” section, and submit your "Accept" recommendation.

Reviewer #1: All comments have been addressed

Reviewer #2: All comments have been addressed

2. Is the manuscript technically sound, and do the data support the conclusions?

Reviewer #1: Yes

Reviewer #2: (No Response)

3. Has the statistical analysis been performed appropriately and rigorously? 

Reviewer #1: Yes

Reviewer #2: (No Response)

4. Have the authors made all data underlying the findings in their manuscript fully available?

Reviewer #1: Yes

Reviewer #2: (No Response)

5. Is the manuscript presented in an intelligible fashion and written in standard English?

Reviewer #1: No

Reviewer #2: (No Response)

6. Review Comments to the Author

Reviewer #1: My comments are added to the attached pdf file. I advise the authors to take their time to correct grammatical and punctuation issues. I tried to point out some of them but it is better to have it proofread by native speakers of English or use online platforms such as grammarly for addressing these issues.

Reviewer #2: (No Response)

7. PLOS authors have the option to publish the peer review history of their article (what does this mean?). If published, this will include your full peer review and any attached files.

Reviewer #1: No

Reviewer #2: No

---

## [Author Response · Author response to Decision Letter 3]

24 Apr 2022

Response to Reviewers Comments

Reviewer #1: My comments are added to the attached pdf file. I advise the authors to take their time to correct grammatical and punctuation issues. I tried to point out some of them but it is better to have it proofread by native speakers of English or use online platforms such as grammarly for addressing these issues.

Following points were rectified as per advice of the respectable Reviewer by considering both MS Word and PDF files while English language corrections were made throughout the text where suggested which can be seen in revised manuscript with track changes. 

MS finalized, where line numbering is now changed due to addition, deletion and inclusion. 

Furthermore, MS have been proofread by native speakers of English Dr. Danish Ibrar and Dr. Shahbaz Khan to whom names are acknowledged.

Moreover, online “Grammarly” source have also been consulted to improve the MS accordingly.

Line 30 NSP corrected 

Line 45 NPS to NSP corrected

Line 68 ‘’due to this physician strongly suggest’’ Sentence removed and modified

Line 91 ‘’Current study engaged several statistical techniques’’ sentence Replaced with ‘’this study used ‘’

Line 92 ‘’ and aiming “replaced with aimed. 

Line 105 comma (,) added 

Line 106 including removed 

Line 107 sentence modified as per suggestion

Line 122 On a plant basis removed and shifted after data after data

Line 129 NP replaced with NPP

Line 128 ‘’after boiling 100 seed’’ sentence removed 

Line 129 Sentence has been revised as cooking time in minutes was recorded after boiling of hard unsoaked seed to soften in distilled water. 

Line 151 Indicating replaced with indicates 

Line 152 Broad sense replaced as broad-sense 

Line 159 Broad sense replaced as broad-sense 

Line 162 Broad sense replaced as broad-sense 

Line 167 Broadened replaced with broaden

Line 166 Comma (,) added after over all

Line 178-180 removed 

Line 184 Whereas has been excluded 

Line 188 NP corrected as NPP

Line 193-194 Removed and embedded

Line 194 ‘’Following’’ deleted 

Line 195 ‘’Broadened’’ has been removed 

Line 205 removed 

Line 211 NP corrected as NPP

Line 129-130 ‘’HSW was positively correlated with BY, in return BY was also positively correlated with 235 PH. Desirable combinations described high potential of the germplasm that can be utilized in 236 future crop improvement programs through selection’’ sentence removed 

Line 231-240 ‘’and much better about evaluated genotypes’’ sentence removed 

Line 240 ‘’Second year’’ added 

Line 246 “Furthermore, scattered plot diagram “changed to ‘’to scatter plot’’

Line 250 Other the genotypes’’ removed 

Line 251 Of the germplasm removed

Line 272 ‘’While on the other hand, at lower left side’’ sentence removed 

Line 285-286 sentence removed 

Line 288 Sentence modified 

Line 295 ‘’as’’ replaced with ‘’ for’’

Line 292 ‘’further more ‘’ corrected 

Line 312 observed to be’’ modified 

Line 327 ‘’Figure-7’’ added 

Line 328 ‘’Found to be’’ was added 

Line 328 ‘’the traits under study’’ removed 

Line 334 It replaces with this

Line 334 Furthermore added 

Line 335 Along with corrected 

Line 341 Table-6 SP corrected as NSP 

Line 341 NP corrected as NPP

Line 344 The removed

Line 344 Scientist replaced with plant breeders

Line 348-349 ‘’It is 356 always to be desirable by breeders to choose high yielding genotypes exhibit highest yield 357 potential to develop varieties or to carry out multi-location trials for varying environments [31] ’’ sentence modified 

Line 356 overall performance’’ sentence modified 

Line 357 to be deleted 

Line 364 ‘’develops’’ sentence modified

Lines 366-367 ‘’There 375 were additionally several environmental factors which influence crop performance as water 376 scarcity affects productivity of legumes at growth stage ‘’ sentence modified

Line 368 are’’ changed to ‘’is’’

Line 370 ‘’present ‘’ deleted 

Line 370 ‘’Genotypes’’ removed 

Line 373 Further’’ removed 

Line 373 ‘as well against‘’ deleted 

Line 379 ‘’during cold season after adopting breeding strategy for lentils 

Line 380 sentence modified and ‘’showed wide variation during both years’’ deleted 

Line 382 ‘’Further more’’ corrected

Line 383 ‘’because’’ added before save energy

Line 383-384 Revised/Rectified 

Line 386 ‘’over all‘’ sentence revised 

Line 387 ‘’further for trials and breeding programs ‘’ revised 

Line 389 Evidenced’’ deleted 

Line 392 ‘’Seed’’ removed 

Line 396 ‘’It’’ replaced with ‘’they’’ 

Line 394 ‘’LP (leaf hairs)’’ corrected 

Line 405 ‘’Further more’’ revised 

Line 406 Understanding of ‘’ revised 

Line 412 ‘’someone should be very careful about sowing time and its stage’’ revised

Line 423- 424 Revised 

Line 427 -428 ‘’therefore, this component is also suggested for further use” revised 

Line 478 genotypic coefficient of variance (GCV), has already mentioned in the table 6

Line 431-572 comprehensively edited as per English language improvement suggestion by respectable Reviewer and expert See file Manuscript with track changes April 24, 2022

Discussion and Conclusion section improved as per suggestion of the expert and their names are included under acknowledgment. See file Manuscript with track changes April 21, 2022

References were rechecked again to remove any discrepancy. 

Figures and supporting material rechecked, revised where fit and data reported in the MS has been counter checked again.

---

## [Decision Letter · Decision Letter 4]

16 May 2022

Estimating genetic variability among diverse lentil collections through novel multivariate techniques

PONE-D-21-20067R4

Dear Dr. Iqbal,

We’re pleased to inform you that your manuscript has been judged scientifically suitable for publication and will be formally accepted for publication once it meets all outstanding technical requirements.

Kind regards,

Tzen-Yuh Chiang

Academic Editor

PLOS ONE

Additional Editor Comments (optional):

Reviewers' comments:

Reviewer's Responses to Questions

**Comments to the Author**

1. If the authors have adequately addressed your comments raised in a previous round of review and you feel that this manuscript is now acceptable for publication, you may indicate that here to bypass the “Comments to the Author” section, enter your conflict of interest statement in the “Confidential to Editor” section, and submit your "Accept" recommendation.

Reviewer #1: All comments have been addressed

2. Is the manuscript technically sound, and do the data support the conclusions?

Reviewer #1: Yes

3. Has the statistical analysis been performed appropriately and rigorously? 

Reviewer #1: Yes

4. Have the authors made all data underlying the findings in their manuscript fully available?

Reviewer #1: Yes

5. Is the manuscript presented in an intelligible fashion and written in standard English?

Reviewer #1: Yes

6. Review Comments to the Author

Reviewer #1: All my previous comments were addressed and the manuscript is in good shape now. I don't have any other comments.

7. PLOS authors have the option to publish the peer review history of their article (what does this mean?). If published, this will include your full peer review and any attached files.

Reviewer #1: No

---

## [Editor Report · Acceptance letter]

14 Jun 2022

PONE-D-21-20067R4 

Estimating genetic variability among diverse lentil collections through novel multivariate techniques 

Dear Dr. Iqbal:

I'm pleased to inform you that your manuscript has been deemed suitable for publication in PLOS ONE. Congratulations! Your manuscript is now with our production department. 

Kind regards, 

on behalf of

Dr. Tzen-Yuh Chiang 

Academic Editor

PLOS ONE